# Hybrid chalcogen bonds in prodrug nanoassemblies provides dual redox-responsivity in the tumor microenvironment

Tian Liu[1], Lingxiao Li[1], Shuo Wang[1], Fudan Dong[1], Shiyi Zuo [1], Jiaxuan Song[1], Xin Wang[1], Qi Lu[1], Helin Wang[1], Haotian Zhang[2], Maosheng Cheng[3], Xiaohong Liu[4], Zhonggui He[1], Bingjun Sun [1]✉ & Jin Sun [1]✉

Sulfur bonds, especially trisulfide bond, have been found to ameliorate the self-assembly stability of homodimeric prodrug nanoassemblies and could trigger the sensitive reduction-responsive release of active drugs. However, the antitumor efficacy of homodimeric prodrug nanoassemblies with single reduction-responsivity may be restricted due to the heterogeneous tumor redox microenvironment. Herein, we replace the middle sulfur atom of trisulfide bond with an oxidizing tellurium atom or selenium atom to construct redox dual-responsive sulfur-tellurium-sulfur and sulfur-selenium-sulfur hybrid chalcogen bonds. The hybrid chalcogen bonds, especially the sulfur-tellurium-sulfur bond, exhibit ultrahigh dual-responsivity to both oxidation and reduction conditions, which could effectively address the heterogeneous tumor microenvironment. Moreover, the hybrid sulfur-tellurium-sulfur bond promotes the self-assembly of homodimeric prodrugs by providing strong intermolecular forces and sufficient steric hindrance. The above advantages of sulfur-tellurium-sulfur bridged homodimeric prodrug nanoassemblies result in the improved antitumor efficacy of docetaxel with satisfactory safety. The exploration of hybrid chalcogen bonds in drug delivery deepened insight into the development of prodrug-based chemotherapy to address tumor redox heterogeneity, thus enriching the design theory of prodrug-based nanomedicines.

Homodimeric prodrug nanoassemblies (HPNAs) are emerging as promising platforms for delivering anticancer drugs[1–11]. By coupling two drug molecules with a suitable linker, the developed homodimeric prodrug could self-assemble into nanoparticles (NPs) without the carriers[1–3]. Compared with traditional nanoparticles such as liposomes and micelles, HPNAs exhibit distinct advantages, such as high drug-loading and low excipient-related toxicity[1,2,12–18]. However, two key factors restrict the application of HPNAs, including poor self-assembling capability and unsatisfactory on-demand target site-bioactivation. Most homodimeric prodrugs show low self-assembly stability because of the imbalanced intermolecular forces, which lead to quick clearance in the systemic circulation[19]. In addition, the homodimeric prodrugs must be converted explicitly into the active form in the target site to realize synergism and attenuation. Rational

[1]Department of Pharmaceutics, Wuya College of Innovation, Shenyang Pharmaceutical University, 110016 Shenyang, People's Republic of China. [2]Department of Pharmacology, School of Life Science and Biopharmaceutics, Shenyang Pharmaceutical University, 110016 Shenyang, People's Republic of China. [3]Key Laboratory of Structure-Based Drug Design and Discovery of Ministry of Education, Shenyang Pharmaceutical University, 110016 Shenyang, People's Republic of China. [4]Department of Pharmaceutics, School of Pharmacy, Shenyang Pharmaceutical University, 110016 Shenyang, People's Republic of China. ✉e-mail: sunbingjun_spy@sina.com; sunjin@syphu.edu.cn

design of HPNAs with good assembly stability and tumor-selective bioactivation is still a significant challenge.

We have previously found that sulfur bond has the nearly 90° bond angle, which could introduce "structural defects" to prevent excessive aggregation of prodrug molecules, thereby facilitating the self-assembly of prodrugs[20–24]. Compared to thioether bond and disulfide bond, trisulfide bond was composed of more sulfur atoms and more sulfur-containing bond angle. Therefore, the trisulfide bond could effectively improve the self-assembly stability, resulting in prolonged blood circulation and enhanced tumor accumulation of HANPs[1]. In addition, the trisulfide bond exhibited ultrahigh reduction sensitivity, which could respond to the high levels of glutathione (GSH) in tumor cells[1]. However, the intracellular microenvironment of tumor cells is characterized by the heterogeneous redox state due to the irregular production of GSH and reactive oxygen species (ROS)[25–28]. Different ROS and GSH levels have been found in different kinds of tumors, even at different times and regions in the same tumor[27,28]. Prodrugs that respond solely to one environmental stimulus might result in limited therapeutic efficacy. Therefore, a new drug delivery system with ultrahigh dual-sensitivity to tumor redox-heterogeneity is required.

In the periodic table of elements, tellurium, selenium, and sulfur belong to the group VIA, also named chalcogen[29]. Compared with sulfur, selenium and tellurium have a larger atomic radius and weaker electron binding capacity, resulting in higher oxidation-responsivity[30–33]. In addition, the bond angles of the selenium bond and tellurium bond are closer to 90° than the sulfur bond, which may further facilitate the self-assembly of homodimeric prodrug[2,20,34–36]. Based on these rationales, we proposed to replace the sulfur atom in the middle of the trisulfide bond with a tellurium atom or selenium atom to construct sulfur-tellurium-sulfur (-STeS-) and sulfur-selenium-sulfur (-SSeS-) hybrid chalcogen bond. It was expected that (i) the introduction of hybrid bonds could further facilitate the self-assembling ability of the homodimeric prodrugs; (ii) the hybrid bond serving as a "double control switch" could be sensitive to both oxidative and reductive stimuli responding to the heterogeneous tumor redox microenvironment due to the unique combination of chalcogens.

In this work, hybrid chalcogen bonds (-STeS- and -SSeS-) bridged HPNAs are developed using docetaxel (DTX) as the model drug (Fig. 1a, b). In addition, -SSS- and -SCS- bridged prodrugs are also synthesized as control. The effects of hybrid chalcogen bonds on the self-assembly, bioactivation, pharmacokinetic behavior, biodistribution, and pharmacodynamics of HPNAs are investigated in detail. Interestingly, the hybrid chalcogen bonds can improve the self-assembly stability of HPNAs, and molecular dynamics simulations illustrate the assembling mechanism. More importantly, -STeS-bonds exhibit ultrahigh dual-responsivity to both oxidation and reduction stimuli, which can effectively respond to the heterogeneous tumor microenvironment.

## Results
### Rational design of DTX homodimeric prodrugs
Four homodimeric DTX prodrugs were designed and synthesized, using -STeS-, -SSeS-, -SSS-, or -SCS- as linkers (Fig. 2a). The corresponding prodrugs were named DTX-STeS-DTX, DTX-SSeS-DTX, DTX-SSS-DTX, and DTX-SCS-DTX, respectively. The synthetic routes of DTX homodimeric prodrugs were illustrated in Supplementary Fig. 1. Briefly, we synthesized the hybrid chalcogen bonds-containing linkers (Supplementary Figs. 2–4). Subsequently, homodimeric DTX prodrugs were synthesized by esterification of linkers and two DTX molecules. The structures of prodrugs were successfully identified by [1]H NMR and high-resolution mass spectrometry (HRMS) (Supplementary Figs. 5–8). The purity of four prodrugs reached over 99% (Supplementary Figs. 5–8).

### Preparation and characterization of HPNAs
According to our previous studies, the linkers played an essential role in driving the self-assembly of prodrugs[1,2,20]. Therefore, we investigated the self-assembly performance of the prodrugs with different linkers. Firstly, the pure HPNAs were prepared via the one-step nano-precipitation, named Non-PEGylated DTX-STeS-DTX NPs, Non-PEGylated DTX-SSeS-DTX NPs, Non-PEGylated DTX-SSS-DTX NPs, and Non-PEGylated DTX-SCS-DTX NPs, respectively. As shown in Supplementary Fig. 9, Non-PEGylated DTX-STeS-DTX NPs exhibited the best colloidal stability, and the particle size remained around 200 nm over 10 days storage. The size of Non-PEGylated DTX-SSeS-DTX NPs became larger on 10 days. In comparison, Non-PEGylated DTX-SCS-DTX NPs showed undesirable stability, with quickly increased particle size in the first 2 h. The results indicated that the slight difference in chemical linkers notably affected the self-assembly ability of the homodimeric prodrugs. The introduction of -SSeS- or -STeS- could effectively enhance the self-assembly stability of the HPNAs.

In order to improve the blood circulation of HPNAs, DSPE-PEG$_{2K}$ (20%, wt.%) was modified onto the surface of the nanoassemblies to prepare PEGylated HPNAs[20,37], which were named DTX-STeS-DTX NPs, DTX-SSeS-DTX NPs, DTX-SSS-DTX NPs, and DTX-SCS-DTX NPs, respectively. The diameters of the four PEGylated HPNAs were ~90 nm (Fig. 2b), the zeta potential was ~−20 mV, and the drug-loading rate was more than 67% (Supplementary Table 1). The prepared nanoassemblies were spherical with uniform size (Fig. 2c). The four PEGylated HPNAs were stable at 4 °C for 35 days (Fig. 2d). Furthermore, the PEGylated HPNAs were diluted with the fetal bovine serum-containing phosphate buffer saline (PBS, pH 7.4). Notably, DTX-SSeS-DTX NPs and DTX-STeS-DTX NPs could remain stable within 24 h, but the sizes of DTX-SSS-DTX NPs and DTX-SCS-DTX NPs were increased (Fig. 2e). These results illustrated that DSPE-PEG$_{2K}$ could improve assembly stability to some extent but did not change the stability difference of HPNAs with different linkers. PEGylated HPNAs were used for the following experiments.

### Self-assembly mechanism of HPNAs
We investigated the self-assembly mechanism of HPNAs using molecular docking and molecular dynamics simulations. The calculated binding energy values were shown as follows (Fig. 3a): DTX-STeS-DTX ($-122.225$ kcal mol$^{-1}$) > DTX-SSeS-DTX ($-120.445$ kcal mol$^{-1}$) > DTX-SSS-DTX ($-119.802$ kcal mol$^{-1}$) > DTX-SCS-DTX ($-119.525$ kcal mol$^{-1}$). From the perspective of thermodynamics, larger binding energy (DTX-STeS-DTX) indicated better self-assembly stability[38]. The aggregation process of HPNAs were shown in Fig. 3b. The compactness and assembling efficiency of nanoassemblies can also be characterized by the radius of gyration. DTX-STeS-DTX self-assembled into a stable cluster with less time spent (9 ns) than that of DTX-SSeS-DTX (10 ns), DTX-SSS-DTX (13 ns) and DTX-SCS-DTX (18 ns), indicating the higher assembly efficiency of DTX-STeS-DTX (Fig. 3c). These results were well consistent with the colloidal stability of HPNAs. Furthermore, according to the preparation process of HPNAs, we added ethanol and DSPE-PEG$_{2K}$ in a corresponding proportion to the dynamic simulation system. In addition, the number of prodrug molecules was increased by up four times ($n = 80$) closer to the studied experiment (Supplementary Fig. 10). Similar to the above results, DTX-STeS-DTX prodrugs were assembled fastest and had the lowest binding energy, followed by DTX-SSeS-DTX prodrugs, DTX-SSS-DTX prodrugs, and finally DTX-SCS-DTX prodrugs (Supplementary Fig. 11 and Supplementary Table 2). The radiuses of gyration of the four systems were shown in Supplementary Fig. 12, and the time length to reach equilibrium was DTX-STeS-DTX < DTX-SSeS-DTX < DTX-SSS-DTX < DTX-SCS-DTX. These results further suggested the best assembly efficiency of DTX-STeS-DTX.

As displayed in Fig. 3a and Supplementary Fig. 13, multiple non-covalent molecular forces drove self-assembly, including π–π stacking

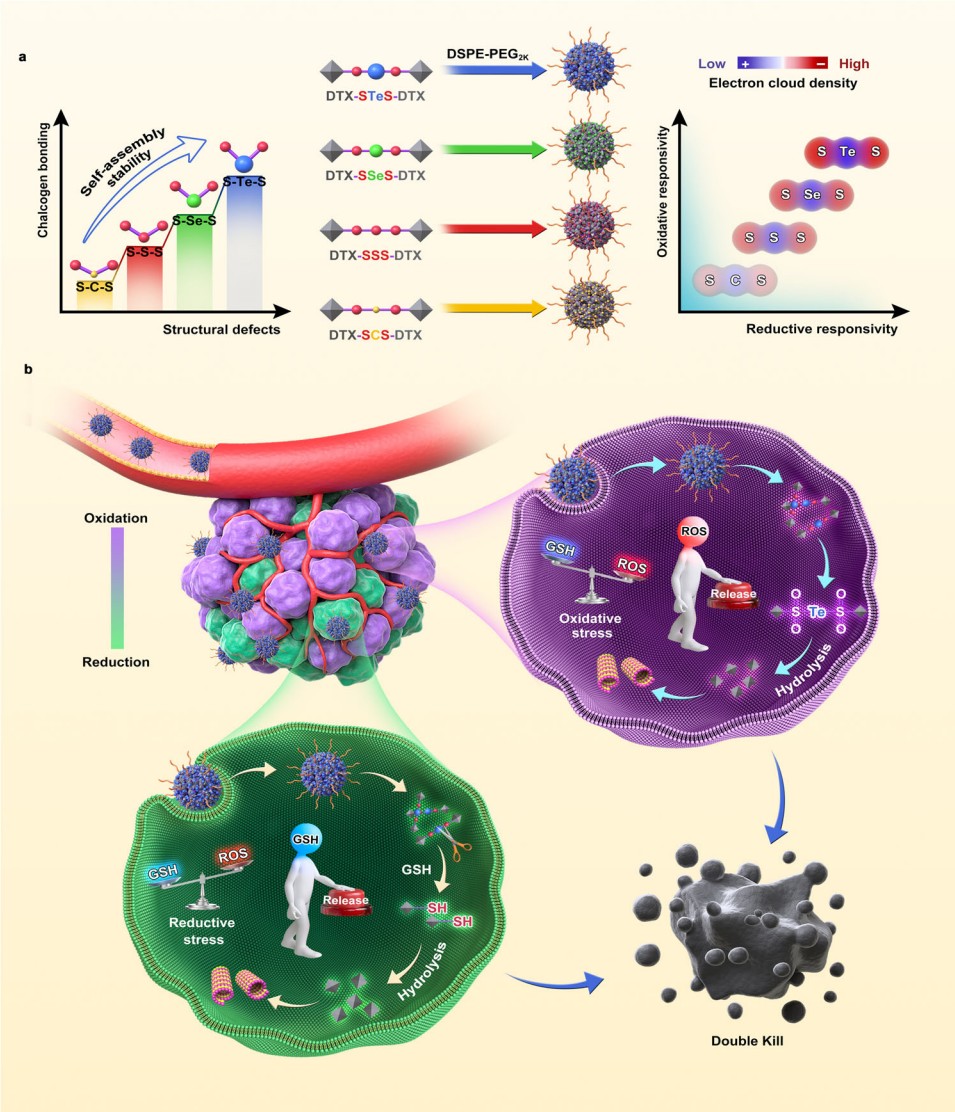

**Fig. 1 | Schematic illustration. a** The self-assembly stability and redox-responsivity of STeS, SSeS, SSS, and SCS bonds. **b** Hybrid chalcogen bond bridged HPNAs with ultrahigh redox dual-responsivity to address tumor heterogeneity.

among DTX molecules, alkyl-alkyl interaction among alkyl part, hydrogen bonding among hydroxyl structure, and chalcogen bonding among linkers. It has been reported that chalcogen bonding has the function of driving self-assembly, and the chalcogen bonding of Te is more robust than that of Se or S[29,39,40]. Therefore, the DTX-STeS-DTX might have stronger self-assembly driving force. In addition, the electrostatic repulsion of DTX-STeS-DTX in the process of aggregation was much weaker than that of other prodrugs (Fig. 3d). These results demonstrated that introducing the tellurium atom or selenium atom could promote the self-assembly of prodrugs by enhanced driving forces.

However, the colloidal stability of HPNAs was based on the balance between aggregation (driving force) and dispersion (steric hindrance)[1,2]. The excessive intermolecular driving force would lead to the over-aggregation of prodrugs and the formation of large particles. Therefore, sufficient steric hindrance was equally necessary for the stable formation of HPNAs. In previous studies, we found that the near 90° vertical sulfur-containing bond angle could provide appropriate steric hindrance, then facilitating the self-assembly of prodrugs[2,20]. Therefore, we calculated the bond angles of -S-Te-S-, -S-Se-S-, -S-S-S-, and -S-C-S- in the corresponding optimized geometry structure. The bond angles were shown in Supplementary Fig. 14: -S-Te-S- (90.94°);

-S-Se-S- (101.88°); -S-S-S- (105.76°), and -S-C-S- (111.62°). Notably, the bond angle of Te in DTX-STeS-DTX prodrug was closer to 90°, which could provide more steric hindrance to balance the intermolecular interaction. Strong intermolecular forces and sufficient steric hindrance provided DTX-STeS-DTX with higher assembly ability.

## Redox dual-responsive bioactivation

To the best of our knowledge, the redox-responsivity of -STeS- and -SSeS- have not been explored previously. The redox dual-responsivity of HPNAs was investigated using hydrogen peroxide ($H_2O_2$, oxidant) and dithiothreitol (DTT, reductant) as redox triggers. At the same time, we investigated the drug release profile of HPNAs in the medium without redox substances. The release rates of HPNAs in 24 h were less than 3% (Supplementary Fig. 15). All HPNAs exhibited oxidation-sensitive drug release profiles, following the order of DTX-STeS-DTX NPs > DTX-SSeS-DTX NPs > DTX-SSS-DTX NPs and DTX-SCS-DTX NPs (Fig. 4a–c and Supplementary Fig. 16). The drug release mechanism of HPNAs has been clarified in Fig. 4d. The prodrugs attacked by $H_2O_2$ firstly formed monoxide, which increased the hydrophilicity of the molecule and accelerated the hydrolysis of adjacent ester bonds. In addition, the monoxides could be further oxidized to multi-oxides, and the chemical structures of the midbodies were detected by mass

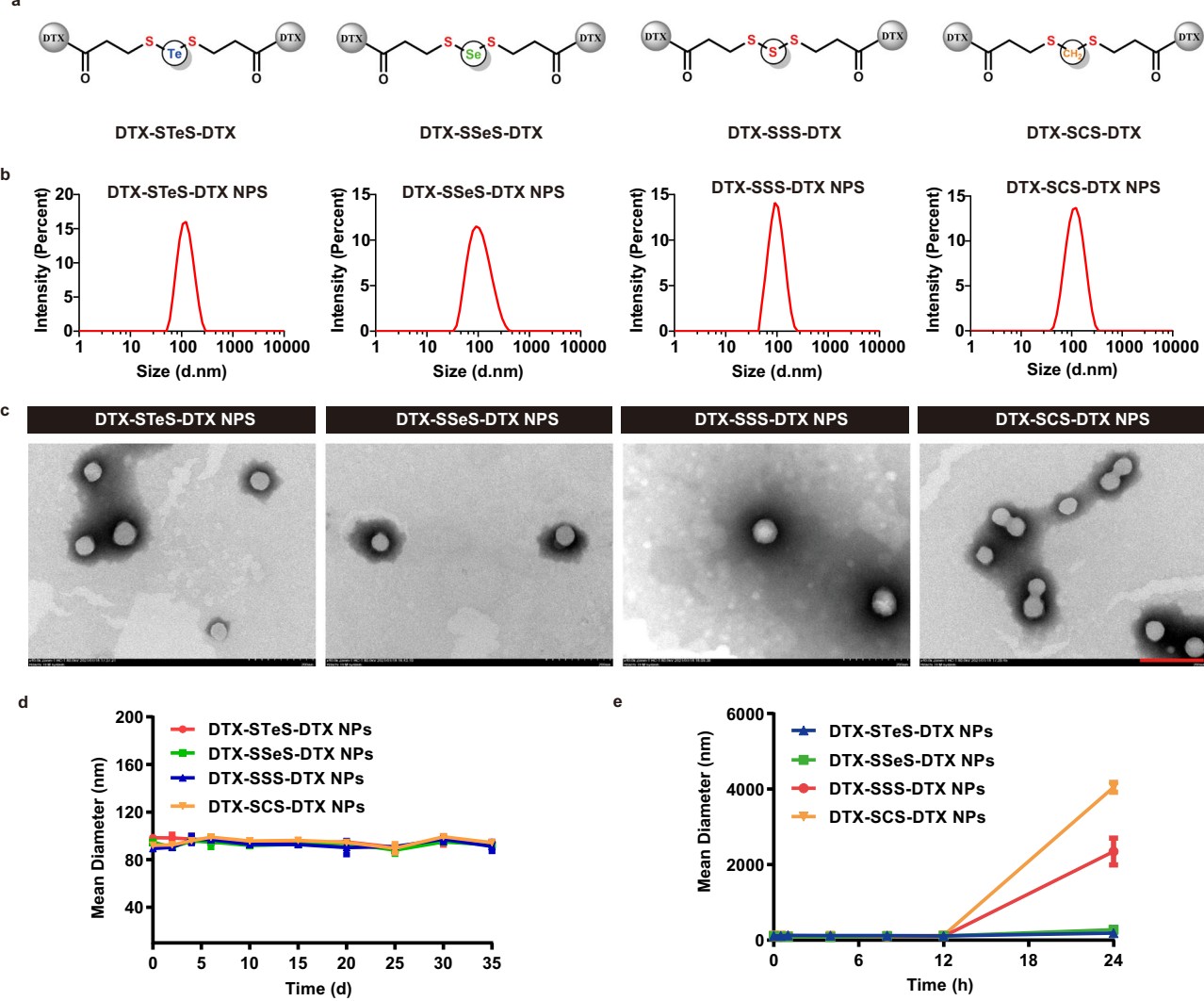

**Fig. 2 | Preparation and characterization of HPNAs. a** Chemical structures of DTX homodimeric prodrugs. **b** Particle size distribution of HPNAs. Source data are provided as a Source Data file. **c** Morphology of HPNAs obtained by TEM. Scale bar represents 200 nm. Experiment was repeated three times independently with similar results. **d** Storage stability of HPNAs at 4 °C. Data are presented as mean ± SD ($n = 3$ independent experiments). Source data are provided as a Source Data file. **e** Particle size change of HPNAs after co-incubation with fetal bovine serum-containing PBS for 24 h. Data are presented as mean ± SD ($n = 3$ independent experiments). Source data are provided as a Source Data file.

spectrometry (Supplementary Figs. 17–20). The oxidation responsivity mainly depended on the electron density of the S atom of SXS bond (X = Te, Se, S, or C), which was closest to the DTX-linked ester bond. For the middle X atoms, the electron-donating effect following the order of Te > Se > S > C. Therefore, the electron cloud was shifted most to the S atom in STeS bond, leading to the highest oxidation-responsivity of STeS bond.

Moreover, we further investigated the intracellular intermediates of prodrugs to verify the release mechanism. The results showed that the intracellular oxidation intermediates of prodrugs were consistent with those in vitro, such as sulfoxide and sulfone (Supplementary Figs. 21–24). In addition, oxidative metabolites such as telluric acid, tellurite acid, selenite acid, and selenite acid were detected in cells treated with DTX-STeS-DTX NPs or DTX-SSeS-DTX NPs (Supplementary Figs. 21–22). These results further proved that the release process of oxidative response existed in cells.

In the presence of DTT, DTX-STeS-DTX NPs, DTX-SSeS-DTX NPs, and DTX-SSS-DTX NPs showed reduction-responsive drug release. The order of reduction-responsivity was described as follows: -STeS- > -SSeS- > -SSS- (Fig. 4e–g). In comparison, minimal DTX

was released from DTX-SCS-DTX NPs at different concentrations of DTT. As oxidation and reduction were reciprocal reactions, it was amazing that DTX-STeS-DTX NPs exhibited ultrahigh redox dual-responsivity. The drug release mechanism of STeS, SSeS, and SSS bond bridged prodrugs in reduction conditions was illustrated in Fig. 4h. Thiolysis evoked by the thiols of DTT promoted the production of thiols-containing prodrug intermediates[41]. These intermediates increased the water solubility of the hydrophobic prodrug, which in turn facilitated hydrolysis of the ester bonds to release DTX. The thiolysis process was mediated by nucleophilic substitution of thiols, so atoms with available unoccupied orbital and lower electron cloud density were more vulnerable to attack. As mentioned above, the electron cloud of STeS was most shifted to S, which caused a lower electron cloud density of tellurium atom. Therefore, the sequence of thiolation rate was as follows: -STeS- > -SSeS- > -SSS-. Carbon atoms lacked available unoccupied orbital to accept electrons, resulting in weak reduction responsiveness. The midbodies were detected by mass spectrometry (Supplementary Figs. 25–27). We also investigated the drug release of HPNAs in the medium with GSH. As

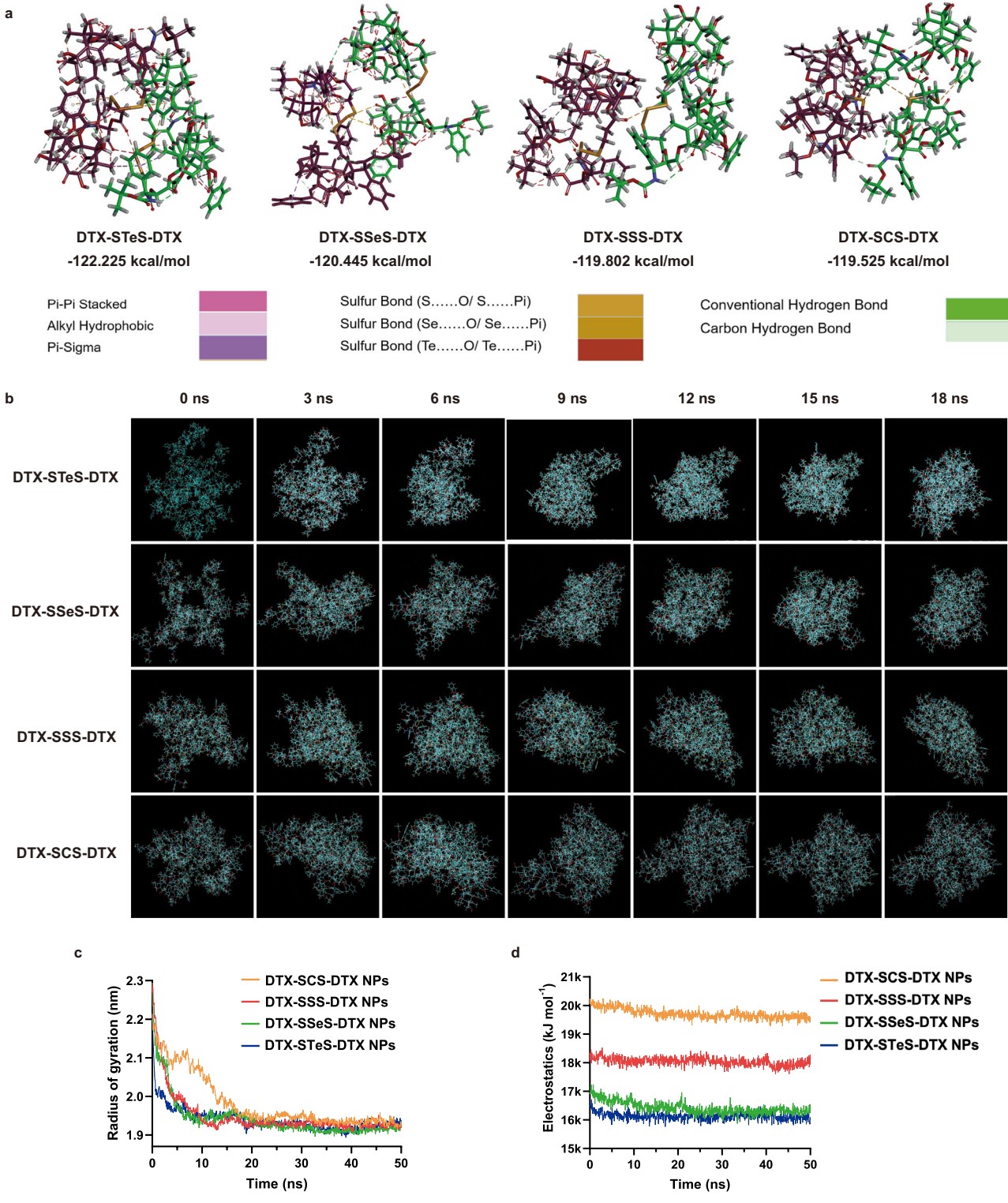

**Fig. 3 | Self-assembly mechanism of HPNAs. a** Intermolecular forces and free energy changes of HPNAs. **b** Schematic diagram of the aggregation process of HPNAs from 0 ns to 18 ns. **c** Changes of the radius of gyration during aggregation. Source data are provided as a Source Data file. **d** Electrostatic repulsion between molecules during aggregation. Source data are provided as a Source Data file.

shown in Supplementary Fig. 28, the reduction-responsivity of HPNAs still followed the order of the DTX-STeS-DTX NPs > DTX-SSeS-DTX NPs > DTX-SSS-DTX NPs > DTX-SCS-DTX NPs, which was in a good agreement with the DTT-trigged drug release.

Similarly, we verified the reduction intermediates in cells. We found that intracellular reducing substances, such as glutathione and

cysteine, were involved in the reduction-responsivity of prodrugs. The intracellular reduction intermediates were shown in Supplementary Figs. 29–31, including DTX-SH, DTX-SXS-GSH, GSH-SXS-GSH, DTX-SXS-Cys, and Cys-SXS-Cys. These results were verified the redox-response release mechanism of HPNAs at the cellular level. Importantly, the STeS bond had the ultrahigh dual-sensitivity to oxidation and

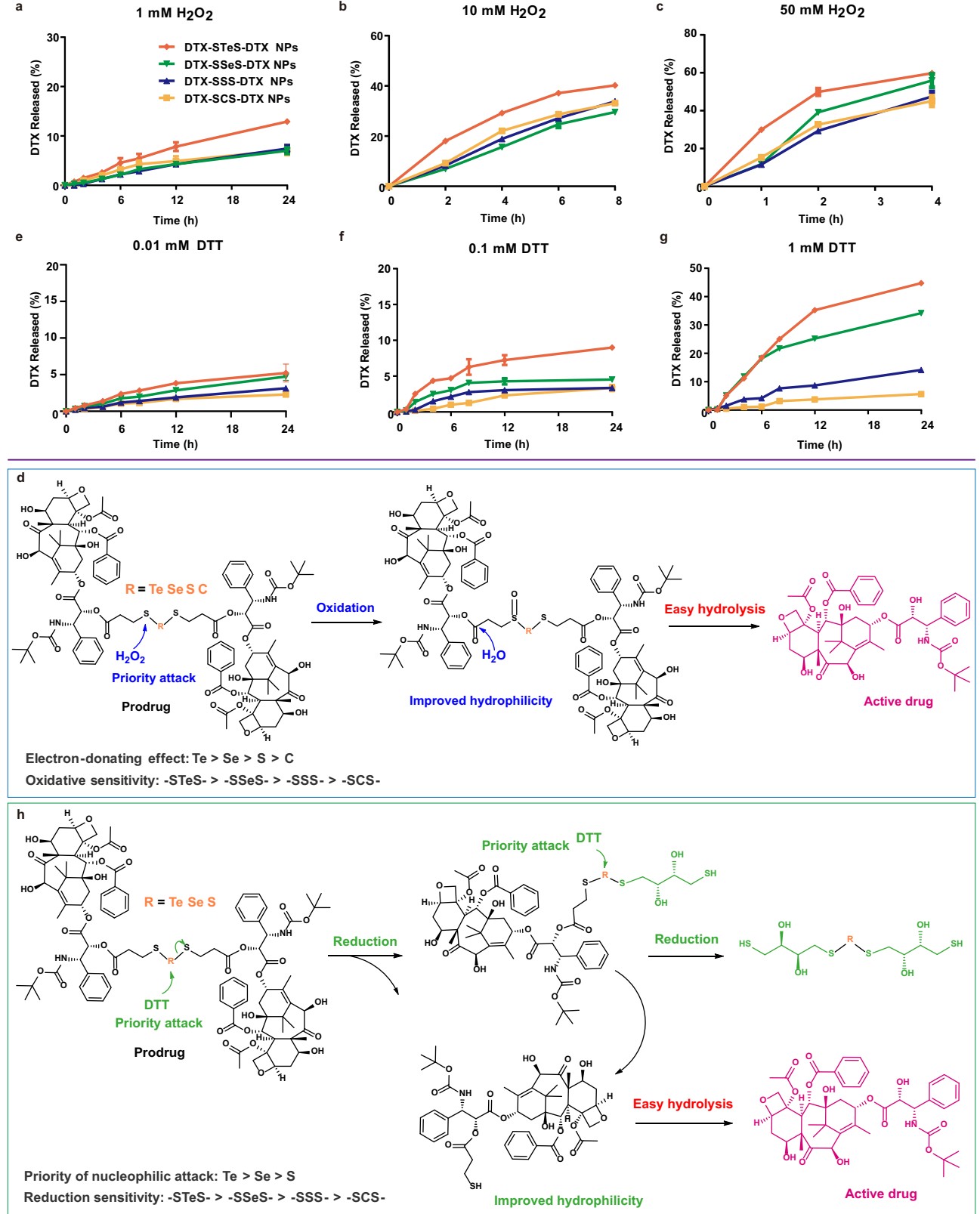

**Fig. 4 | Redox dual-responsive bioactivation. a** 1 mM $H_2O_2$; **b** 10 mM $H_2O_2$; **c** 50 mM $H_2O_2$; **e** 0.01 mM DTT; **f** 0.1 mM DTT; **g** 1 mM DTT. All data of **a**–**c**, **e**–**g** were presented as the mean ± SD ($n = 3$ independent experiments).

Source data are provided as a Source Data file. Mechanisms of the oxidation-responsiveness (**d**) and reduction-responsiveness (**h**) of HPNAs.

reduction, suggesting that it could effectively respond to the heterogeneous tumor microenvironment.

## Cytotoxicity and intracellular bioactivation

The cytotoxicity of HPNAs was investigated in tumor cells (4T1 cells, Hepa 1–6 cells, and B16F10 cells) and normal cells (3T3 cells) (Supplementary Fig. 32). Compared with the normal cells (3T3 cells), the levels of both ROS and GSH in tumor cells (4T1, Hepa 1–6, and B16F10 cells) were higher (Supplementary Figs. 33a–c and 34). The hybrid bond HPNAs with redox dual-responsiveness could effectively respond to the heterogeneous tumor microenvironment. As the results show, the HPNAs exhibited weaker cytotoxicity than Taxotere due to the delayed drug release process, following the order of DTX-STeS-DTX NPs > DTX-SSeS-DTX  NPs > DTX-SSS-DTX  NPs > DTX-SCS-DTX  NPs (Fig. 5a–d and Supplementary Table 3). In addition, the four HPNAs showed similar cellular uptake, suggesting the difference in cytotoxicity may be due to the release rate of active DTX (Fig. 5e, f and Supplementary Fig. 35). Hence, the intracellular release of DTX from HPNAs was investigated, and the results were consistent with the results of cytotoxicity (Fig. 5g–h and Supplementary Fig. 36). DTX-STeS-DTX NPs, with ultrahigh redox dual-responsivity, showed the fastest DTX release rate and the highest tumor cytotoxicity. In addition, compared with other HPNAs, DTX-STeS-DTX NPs showed the strongest inhibitory effect on tubulin and elicited the most extensive apoptosis, which was also in an agreement with the intracellular release (Fig. 5i and Supplementary Figs. 37, 38 and 39). The apoptosis rates of Taxotere and HPNAs were as follows: Taxotere (52.2%) > DTX-STeS-DTX NPs (43.15%) > DTX-SSeS-DTX NPs (37.04%) > DTX-SSS-DTX NPs (32.3%) > DTX-SCS-DTX NPs (25.8%). Moreover, compared with Taxotere, HPNAs exhibited a higher selectivity index among tumor and normal cells (Supplementary Table 4). This result demonstrated that the redox-responsivity of prodrugs improved the tumor-selective bioactivation of HPNAs, which could also reduce the systemic toxicity of Taxotere.

Sulfur/selenium/tellurium-containing HPNAs can affect the redox equilibrium of tumor cells while undergoing redox-response[42,43]. It has been found that tumor cells are more sensitive to the redox equilibrium than normal cells[26,44]. Thus, changes in ROS levels and GSH/GSSG ratio in 4T1 cells after HPNAs treatment were examined. Compared with the control group, DTX-STeS-DTX NPs, DTX-SSeS-DTX NPs, and DTX-SSS-DTX NPs significantly increased intracellular ROS levels and decreased GSH/GSSG (Supplementary Figs. 40–41). These results indicated an increased level of intracellular oxidation, which would further accelerate drug release and tumor cell apoptosis.

## Pharmacokinetics and biodistribution of HPNAs

The pharmacokinetic behavior of HPNAs was influenced by self-assembly stability and chemical stability. As shown in Supplementary Fig. 42, the chemical stability of HPNAs in rat plasma was consistent with the results of redox-sensitivity release. DTX-SCS-DTX NPs exhibited good chemical stability for 24 h (over 80% kept intact), followed by DTX-SSS-DTX NPs (nearly 60%), DTX-SSeS-DTX NPs (nearly 20%), and DTX-STeS-DTX NPs (nearly 20%).

The pharmacokinetic profiles of the prodrugs, the released DTX, and the sum were described in Fig. 6a–c and Supplementary Table 5. DTX in Taxotere was rapidly cleared from the blood (Fig. 6a). By contrast, HPNAs alleviated the blood clearance of DTX. The area under the curve (AUC) of DTX-STeS-DTX NPs, DTX-SSeS-DTX NPs, DTX-SSS-DTX NPs, DTX-SCS-DTX NPs was 31.98-, 31.61-, 31.85-, and 127.35-fold higher than Taxotere, respectively (Supplementary Table 5). The amount of released DTX in blood was consistent with the chemical stability of the HPNAs (Fig. 6b). More DTX was released from DTX-STeS-DTX NPs and DTX-SSeS-DTX NPs than those from other HNPAs, possibly because of the higher redox-responsivity of

hybrid chalcogen bond. Despite the relatively poor chemical stability of DTX-STeS-DTX NPs and DTX-SSeS-DTX NPs, there was no statistical difference in the AUC values of DTX-STeS-DTX NPs, DTX-SSeS-DTX NPs, and DTX-SSS-DTX NPs (Supplementary Fig. 43). This result suggested that the good self-assembly stability partially compensated for the poor chemical stability of DTX-STeS-DTX NPs and DTX-SSeS-DTX NPs.

DiR-labeled HPNAs were used to investigate the biodistribution in 4T1 tumor-bearing BALB/c mice. DiR-labeled HPNAs showed much higher fluorescent signals in tumors than DiR solution (Fig. 6d–f). These results proved that HPNAs could effectively accumulate in tumors. However, the accumulation of DiR-labeled HPNAs instead of active DTX in the tumor couldn't directly reflect the end-point treatment outcomes. Therefore, we further investigated the level of DTX in tumor tissue using LC-MS. As shown in Fig. 6g, the accumulation of Taxotere in tumor was gradually decreased over time. The rapid distribution of Taxotere was also consistent with its pharmacokinetic characteristics. In comparison, the released DTX of HPNAs in the tumor was gradually increased. At 24 h, DTX-STeS-DTX NPs, with ultrahigh redox dual-responsivity, exhibited significantly higher DTX concentration in tumors than the other groups.

## Antitumor efficacy of HPNAs

The antitumor efficacy of HPNAs and Taxotere on 4T1 tumor were evaluated at different doses. At low-dose (2 mg kg$^{-1}$), the groups of Taxotere and DTX-SCS-DTX NPs showed weak antitumor effects (Fig. 7a–d). In contrast, DTX-SSS-DTX NPs, DTX-SSeS-DTX NPs, and DTX-STeS-DTX NPs exhibited more effective tumor growth inhibition, with fewer metastatic nodules in the lung (Supplementary Fig. 44). Subsequently, the antitumor effects of HPNAs at moderate-dose (10 mg kg$^{-1}$) were investigated. DTX-SCS-DTX NPs, DTX-SSS-DTX NPs, and DTX-SSeS-DTX NPs groups slowed the tumor growth to a certain extent (Fig. 7e–h). In comparison, Taxotere and DTX-STeS-DTX NPs showed higher antitumor efficacy. Especially, significant cellular apoptosis of tumor induced by the DTX-STeS-DTX NPs (2 mg kg$^{-1}$ and 10 mg kg$^{-1}$) was confirmed by TUNEL staining (Supplementary Figs. 45 and 46). In addition, DTX-STeS-DTX NPs decreased the level of Ki-67 in tumor sections, suggesting effective suppression of tumor proliferation (Supplementary Figs. 45 and 46).

To explore the therapeutic effect of DTX-STeS-DTX NPs at the tolerated dose, we further increased the dosage to 15 mg kg$^{-1}$ and 20 mg kg$^{-1}$. The antitumor effect of the DTX-STeS-DTX NPs at 15 mg kg$^{-1}$ was superior to Taxotere at the same dose and was optimal at 20 mg kg$^{-1}$, with no effect on body weight (Fig. 7i–l).

In terms of safety, Taxotere caused serious systemic toxicity, and the mice suffered weight loss even at low doses (2 mg kg$^{-1}$) (Fig. 7b). Moreover, decreases in white blood cells and lymphocytes in the Taxotere group at 10 mg kg$^{-1}$ suggested bone marrow suppression (Supplementary Fig. 47). Nucleolus pyknosis of liver slice and increases in alanine aminotransferase (ALT) and aspartate aminotransferase (AST) levels occurred in the Taxotere group, indicating liver injury (Supplementary Figs. 47 and 48). In comparison, the mice treated with HPNAs remained normal weight (Fig. 7b, f, j). Furthermore, hematological parameters, blood analysis, and H&E staining of HPNAs have no indication of toxicity and histological changes, even in the group of DTX-STeS-DTX NPs at 15 mg kg$^{-1}$ (Supplementary Figs. 44 and 47–50). These results indicated that DTX-STeS-DTX NPs not only had potent antitumor efficacy but also showed good safety. Therefore, the introduction of STeS hybrid chalcogen bond overcame the two limitations of HPNAs: (i) STeS bonds could improve the self-assembly stability of HPNAs for long blood circulation and high tumor accumulation; and (ii) the ultrahigh redox dual-sensitivity of STeS bond could respond to the heterogeneous tumor redox microenvironment, enabling sufficient drug bioactivation in tumor cells.

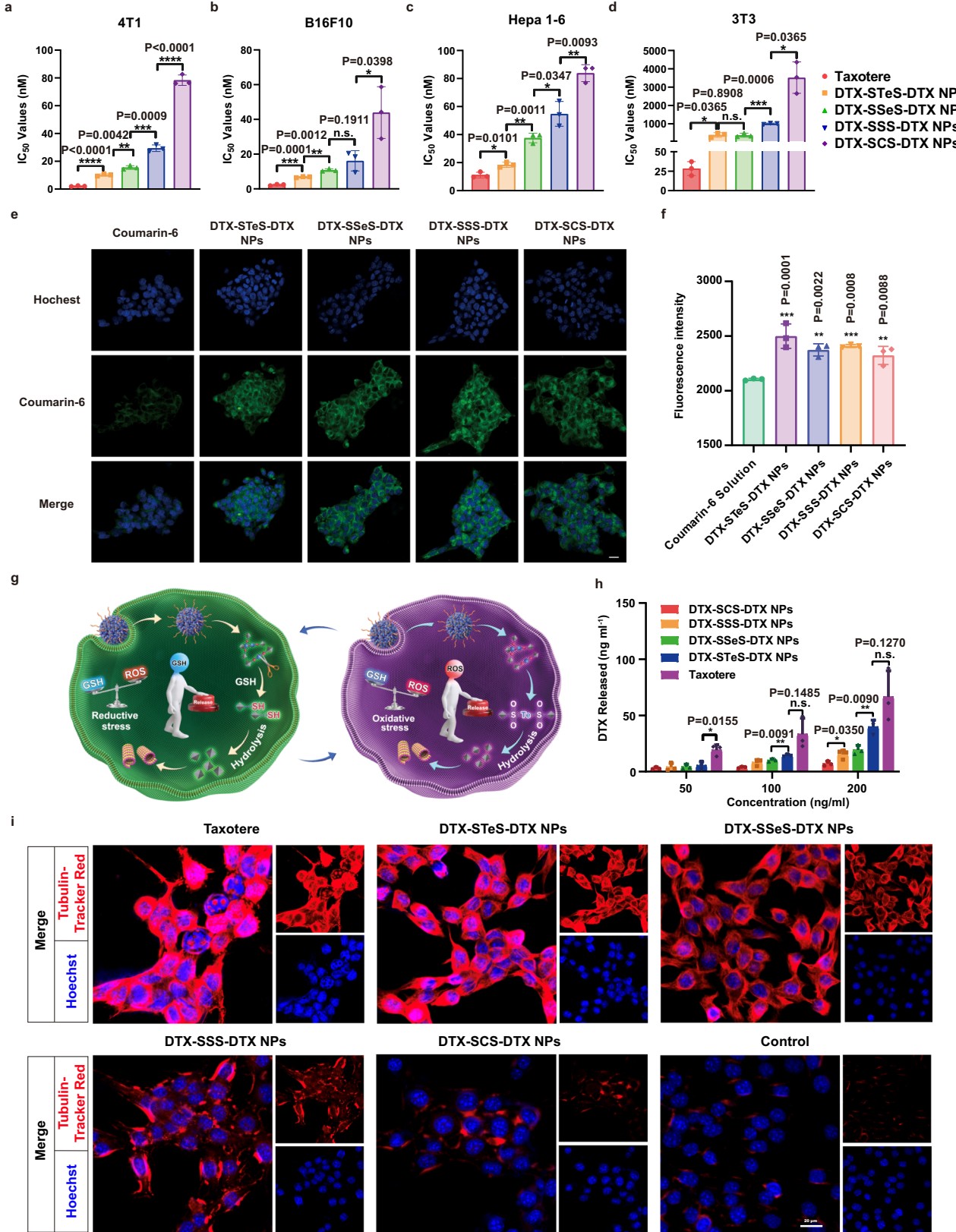

## Discussion

Homodimeric prodrug nanoassemblies are promising nano-strategies to improve the efficacy and safety of chemotherapeutical nanomedicines. The linker plays a crucial role in both self-assembly stability and tumor-selective bioactivation. This study constructed ultrahigh redox dual-responsivity -STeS- and -SSeS- hybrid chalcogen bonds.

The hybrid chalcogen bonds effectively promoted the self-assembly of the homodimeric prodrug by providing strong intermolecular force (chalcogen bonding) and steric hindrance (nearly 90° bond angle). The improved colloid stability of HPNAs guaranteed prolonged systemic circulation and high tumor accumulation. Moreover, the hybrid chalcogen bonds, especially -STeS- bond, serve as "a double control

**Fig. 5 | Cytotoxicity and intracellular bioactivation. a–d** IC$_{50}$ values of Taxotere and HPNAs in 4T1 cells, B16F10 cells, Hepa 1–6 cells, and 3T3 cells. All data are presented as mean ± SD ($n = 3$ independent experiments). n.s. (no significance) $P > 0.05$, *$P < 0.05$, **$P < 0.01$, ***$P < 0.001$ and ****$P < 0.0001$ by two-tailed Student's $t$-test. Source data are provided as a Source Data file. **e** Cellular uptake of free coumarin-6 or coumarin-6-labeled HPNAs at 2 h. Scale bar represents 10 µm. Experiment was repeated twice independently with similar results. **f** Results of cell uptake by flow cytometry. Data are presented as mean ± SD ($n = 3$ independent experiments). One-way ANOVA (one-sided) with Dunnett's multiple comparisons test was used for the analysis of data and adjusted $P$-value. Source data are provided as a Source Data file. **g** Illustration of intracellular drug release. **h** Free DTX released from HPNAs after incubation with 4T1 cells for 72 h. Data are presented as mean ± SD ($n = 3$ independent experiments). n.s. (no significance) $P > 0.05$, *$P < 0.05$ and **$P < 0.01$ by two-tailed Student's $t$-test. Source data are provided as a Source Data file. **i** Inhibitory effect of Taxotere and HPNAs on tubulin. Scale bar represents 20 µm. Experiment was repeated twice independently with similar results.

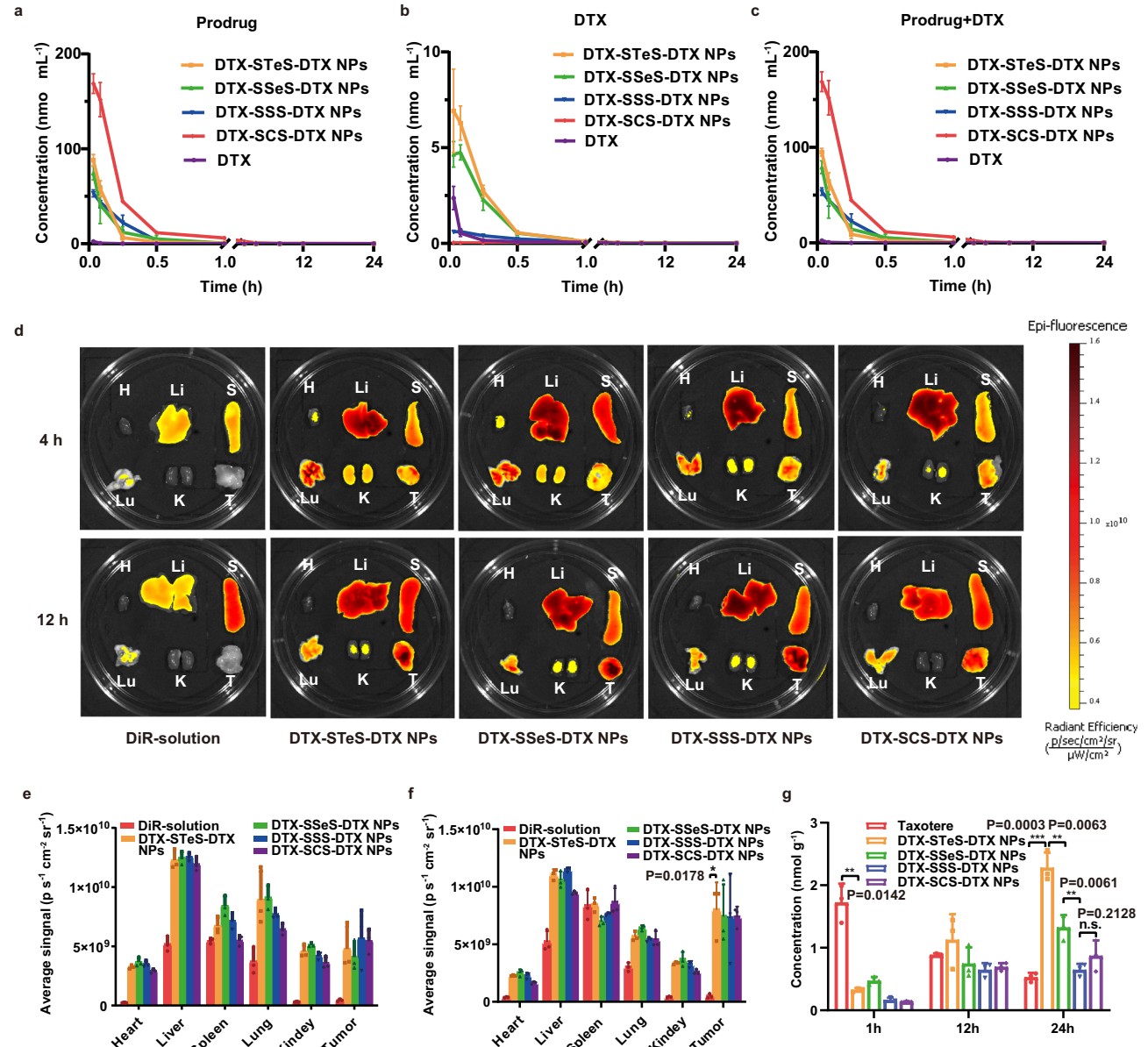

**Fig. 6 | Pharmacokinetics and biodistribution of HPNAs.** Pharmacokinetic profiles of the (**a**) prodrugs, (**b**) released DTX, and (**c**) total equivalent DTX. All data of **a**–**c** were presented as mean ± SD ($n = 3$ rats). Source data are provided as a Source Data file. **d** Biodistribution of DiR-solution or DiR-labeled HPNAs. Quantitative analysis of biodistribution at 4 h (**e**) and 12 h (**f**). **g** The accumulated DTX of Taxotere and HPNAs in tumor. All data of **e**–**g** were presented as mean ± SD ($n = 3$ mice). n.s (no significance) $P > 0.05$, *$P < 0.05$, **$P < 0.01$, and ***$P < 0.001$ by two-tailed Student's $t$-test. Source data are provided as a Source Data file.

switch" to respond to the heterogeneous tumor redox microenvironment. Accordingly, DTX-STeS-DTX NPs with optimal self-assembly stability and ultrahigh redox dual-responsivity exhibited an ideal antitumor effect and safety. These advances provided a promising strategy for overcoming the dilemma of tumor heterogeneity in terms of rational design prodrug-based nanomedicines.

According to our knowledge, previous literature reports on redox-responsive prodrugs have used chemical linker containing a single element (sulfur, selenium, or tellurium). This study used the concept of hybrid chalcogen bonds in the field of drug delivery and tumor treatment, as well as demonstrated the potential of hybrid bonds for on-demand bioactivation as redox dual-responsive

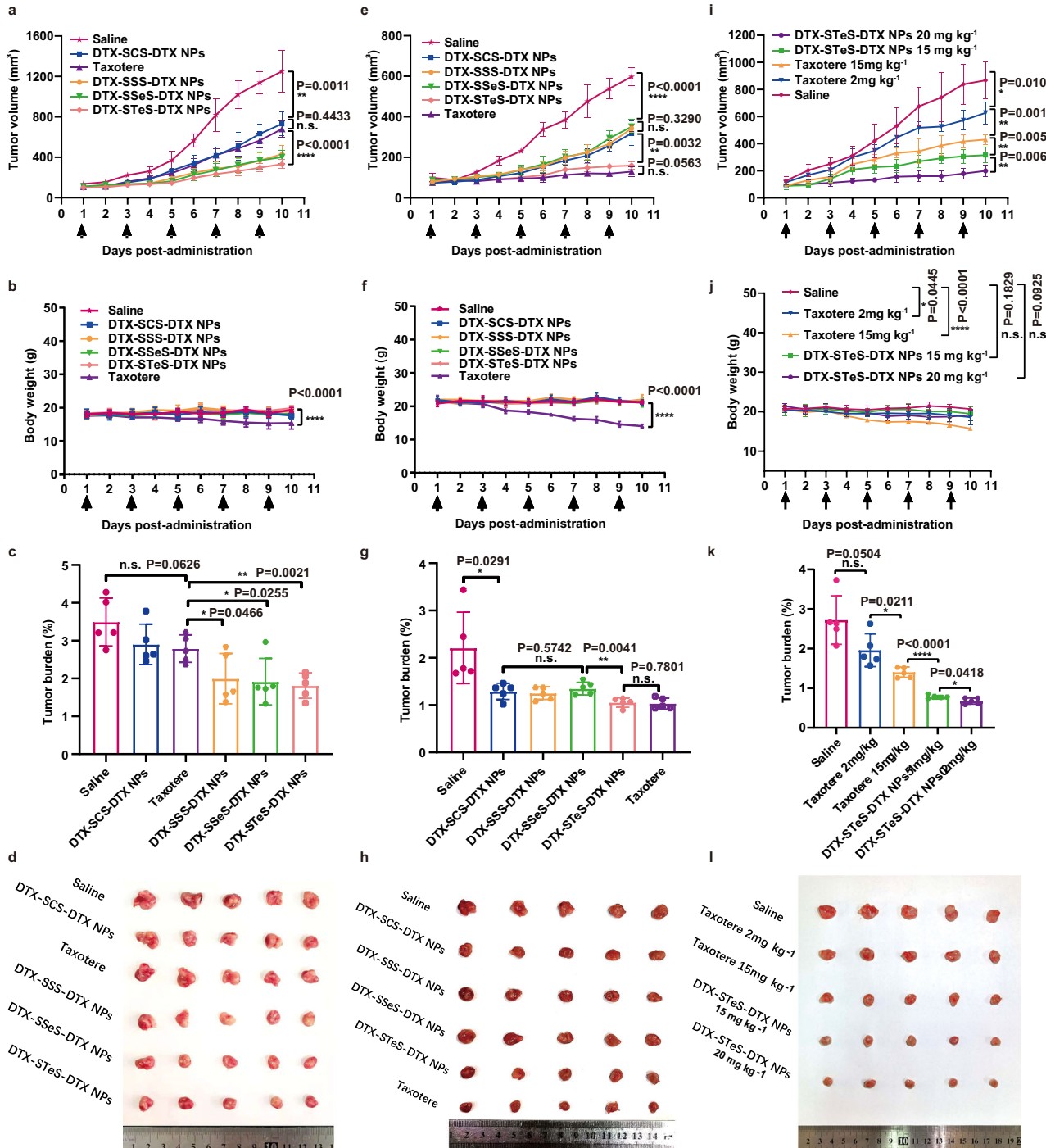

**Fig. 7 | Antitumor efficacy of HPNAs.** Tumor growth profiles, body weight, tumor burden, and tumor photos: **a**–**d** low-dose (2 mg kg⁻¹), **e**–**h** moderate-dose (10 mg kg⁻¹), and **i**–**l** tolerated dose (15 mg kg⁻¹ and 20 mg kg⁻¹). All data of **a**–**c**, **e**–**g**, **i**–**k** were presented as the mean ± SD ($n = 5$ mice). n.s (no significance) $P > 0.05$, *$P < 0.05$, **$P < 0.01$, ***$P < 0.001$, and ****$P < 0.0001$ by two-tailed Student's $t$-test. Source data are provided as a Source Data file.

component. This study was expected to provide a potential development strategy and theoretical basis for prodrug-based antitumor chemotherapy.

In the process of the basic idea to the clinical trial, HPNAs suitable for industrial production and toxicity related to selenium and tellurium need to be paid much attention. Specifically, the final administration form of HPNAs is necessary to be determined (lyophilized powder or solution). In addition, the chemical and physical stability need to meet the requirements of production and sterilization. Moreover, it might be necessary to closely

monitor the toxicity related to selenium and tellurium elements in clinical trials.

## Methods

### Ethical statement

Our research complies with all relevant ethical regulations. All the animal protocols were performed in line with the Guidelines for the Care and Use of Laboratory Animals and approved by the Institutional Animal Ethical Care Committee (IAEC) of Shenyang Pharmaceutical University.

## Materials

Sodium tellurite, Sodium selenite, Sodium thiosulfate, Sodium sulfide, $H_2O_2$, 3-Mercaptopropionic acid, 3-Bromopropionic acid, 4-Dimethylaminopyrideine (DMAP), 1-Ethyl-3-(3-dimethyllaminopropyl) carbodiimide hydrochloride (EDCI), and coumarin-6 were obtained from Aladdin Biochemical Technology Co. Ltd. (Shanghai, China). S,S′-Methylenebis(3-mercaptopropionic acid) were obtained from Sigma-Aldrich (Shanghai) Trade Co. Ltd. (Shanghai, China) Docetaxel, DL-Dithiothreitol (DTT), Glutathione (GSH), 1,1′- Dioctadecyl-3,3,3′,3′-tetra-methyindotricarbocyanine iodide (DiR), 3-(4,5-dimethylthiazol-2-yl)-2,5-diphenyl tetrazolium bromide (MTT), Dulbecco's modified eagle medium (DMEM), Roswell Park Memorial Institute (RPMI−1640), and trypsin were obtained from Meilun Biotechnology Co. Ltd. (Dalian, China). 2-distearoyl-sn-glycerol-3-phosphoethanolamine-N-methyl (polyethylene glycol)-2000 (DSPE-PEG$_{2k}$) were purchased from Shanghai Advanced Vehicle Technology Pharmaceutical Co., Ltd. (Shanghai, China). Hoechst 33342 was obtained from BD Biosciences (USA). Ki67 cell proliferation kit and TUNEL apoptosis assay kit were purchased from Service Biotech Co., Ltd. (China). DAPI was purchased from Vector laboratories (Burlingame). The other reagents were of analytical or high-performance liquid chromatography (HPLC) grade.

## Cell lines and animals

4T1 cells (Serial: TCM32), B16F10 cells (Serial: TCM36) and 3T3 cells (Serial: GNM25) were obtained from the Cell Bank of Type Culture Collection of Chinese Academy of Sciences (Shanghai, China). Hepa 1−6 cells were obtained from Beijing Dingguo Changsheng Biotechnology Co., Ltd. Cell lines validation using short tandem repeat (STR) markers were performed by GENTIC TESTING BIOTECHNOLOGY Co., Ltd. (Jiangsu, China). In detail, eighteen STR loci were amplified using multiplex PCR. One additional marker (Human TH01) was used to screen for the presence of human species. The cell line sample was processed using the ABI Prism 3130 XL Genetic Analyzer. Data were analyzed using Gene Mapper ID 3.2 software (Applied Biosystems). Appropriate positive and negative controls were run and confirmed for each sample submitted. All cell lines tested negative for mycoplasma contamination.

Sprague-Dawley rats (male, 6 weeks old) and BALB/c mice (female, 11 weeks old) were supplied by the Animal Center of Shenyang Pharmaceutical University (Shenyang, Liaoning, China). The living environment of animals were maintained at a temperature of ~25 °C with a 12 h light/dark cycle, with free access to standard food and water. The humane endpoints included tumor burden exceeding 5% of normal body weight, animal weight loss exceeds 20% of normal animal weight, ulcer at tumor growth point, and sustained self-mutilation in animals. The humane end-point was approved by Certification and Accreditation Administration of the People's Republic of China (CNCA). Cervical dislocation under deep anesthesia was adopted for euthanasia. All animal procedures were carried out under the guidelines approved by the Institutional Animal Care and Use Committee (IACUC) of Shenyang Pharmaceutical University.

## Synthesis of 3,3′-(telluriumdithio)-dipropionic acid

The synthesis followed a reported method as shown in Supplementary Fig. 1a[45]. Sodium tellurite (0.222 g, 1 mmol) and mercaptopropionic acid (0.5228 mL, 6 mmol) were dispersed with water. While stirring, the sodium tellurite solution was added dropwise into the mercaptopropionic acid solution through a liquid adding funnel. Stir for 5 min at room temperature to precipitate golden yellow crystals. Vacuum drying at 25 °C to obtain golden yellow solid (80% yield). The structure was determined by $^1$H NMR spectroscopy (400 MHz, Bruker AV-400). The NMR data were analyzed by MestReC 4.9.9.9 software.

## Synthesis of 3,3′-(selenodithio)-dipropionic acid

3,3′-(selenodithio)-dipropionic acid was synthesized followed a reported method as shown in Supplementary Fig. 1b[46]. Sodium selenite

(172.9 mg, 1 mmol) and mercaptopropionic acid (348.6 µL, 4 mmol) were dispersed in 25 mL of dilute sulfuric acid, respectively. Under stirring, the dilute sulfuric acid solution of sodium selenite was dripped into the dilute sulfuric acid solution of mercaptopropionic acid through a liquid adding funnel. After 1 h (25 °C), the white precipitate was precipitated. Subsequently, 100 mL of dilute sulfuric acid solution was added and stirred at 70 °C for 40 min until the precipitate dissolved. It was cooling to room temperature and precipitating white crystals. Vacuum drying at 25 °C to obtain a milky white solid (90% yield). The structure was confirmed by $^1$H NMR spectroscopy (400 MHz, Bruker AV-400). The NMR data were analyzed by MestReC 4.9.9.9 software.

## Synthesis of 3,3′-trithiodipropionic acid

3,3′-Trithiodipropionic acid was synthesized, referring to a previously reported method as shown in Supplementary Fig. 1c[1]. Sodium thiosulfate (10.67 g, 43 mmol) and 3-bromopropionic acid (5.05 g, 33 mmol) were respectively dispersed with water. While stirring at 50−60 °C, drop the 3-bromopropionic acid solution into the sodium thiosulfate solution and react for 4−5 h. After cooling to 25 °C, sodium sulfide nonahydrate (2.40 g, 10 mmol) aqueous solution was added to the system for 12 h. The reaction solution was extracted with ethyl acetate to obtain white solid (30% yield). The structure was confirmed by $^1$H NMR spectroscopy (600 MHz, Bruker AV-600). The NMR data were analyzed by MestReC 4.9.9.9 software.

## Synthesis of DTX homodimeric prodrugs

DTX (403.95 mg, 0.5 mmol), diacid linker (3,3′-(telluriumdithio)-dipropionic acid, 3,3′-(selenodithio)-dipropionic acid, 3,3′-Trithiodipropionic acid, or S,S′-Methylenebis(3-mercaptopropionic acid), 0.25 mmol), EDCI (191.7 mg, 1 mmol) and DMAP (6.11 mg, 0.05 mmol) was dissolved in anhydrous dichloromethane as shown in Supplementary Fig. 1d. Then the reaction system was stirred for 12 h under nitrogen at 25 °C. Then, another EDCI (95.85 mg, 0.5 mmol) and DMAP (6.11 mg, 0.05 mmol) were added into reaction system for 12 h at 25 °C. The product was purified by reversed-phase HPLC. The prodrugs were confirmed by a fourier transform ion cyclotron resonance mass spectrometer (FT-MS, solariX, Bruker, Germany) and $^1$H NMR spectral analyses (600 MHz, Bruker AV-600, Germany). solariX was used to identify the molecular weight of synthetic prodrugs. The type of mass analyzer is the Fourier Transform Ion Cyclotron Resonance (FT-ICR). MS acquisition settings adopted Acquisition Mode: Single MS; Polarity: Positive; Source Accumulation: 0.000 sec; Ion Accumulation Time: 0.280 s; Laser Power: 20.0 lp; Laser Shot Frequency: 0.001 s; and Apodization: Full-Sine. High-resolution mass spectrums were acquired by software ftmsControl 2.1 and analyzed by Bruker Compass DataAnalysis 4.4. The validation results were credible when the error between the measured and theoretical molecular weights was not more than 5 ppm. The NMR data were analyzed by MestReC 4.9.9.9 software. The purity was determined by a reverse-phase HPLC system.

## Preparation and characterization of DTX homodimeric HPNAs

The one-step nano-precipitation prepared the DTX homodimeric HPNAs. In brief, 8 mg DTX homodimeric prodrugs and 2 mg DSPE-PEG$_{2K}$ (20%, w/w) were dissolved in 0.4 mL ethanol. Under stirring, the mixture solution was added dropwise into 4 mL deionized water. The ethanol in the HPNAs was removed using a rotary evaporator at 30 °C. The non-PEGylated HPNAs were prepared according to the same procedure without DSPE-PEG$_{2K}$. The coumarin-6/DiR-labeled HPNAs were prepared by co-assembling prodrugs with coumarin-6/DiR. The particle size and surface potential of the HPNAs were measured by a Zetasizer (Malvern Co., UK) using Zetasizer Software 7.01. The morphologies of the HPNAs were observed by transmission electron microscopy (TEM, Hitachi, Japan) using 0.5% phosphotungstic acid for

negative staining. The colloidal stability of non-PEGylated HPNAs ($0.1\,mg\,mL^{-1}$) stored at $25\,°C$ was investigated by monitoring the particle size for 240 h. In addition, Zetasizer (Malvern, UK) was used to measure the stability of HPNAs after being stored at $4\,°C$ for 6 days. Further, the colloidal stability of HPNAs in PBS ($v/v = 1:20$) containing 10% FBS was analyzed using dynamic light scattering.

## Self-assembly mechanism of HPNAs

The small molecules involved in the calculation project of bond angle are optimized by xtb software in gnf2 and gbsa implicit water model[47]. The molecular structure was searched by Molclus 1.9.9 software, and 10 possible configurations were obtained[48]. On this basis, xtb software was combined to optimize and calculate the binding energy[47]. In addition, we performed molecular dynamics (MD) simulations. The four monomer prodrug molecules (DTX-STeS-DTX, DTX-SSeS-DTX, DTX-SSS-DTX, and DTX-SCS-DTX) were constructed using the Gauss-View 5 software. The monomers were optimized using the PM6 method based on the Gaussian 09 procedure. After the stable optimal structures were calculated, four groups of composite systems consisting of 20 monomers were constructed using Packmol program, and then molecular dynamics simulation was performed for the system. The molecular RESP charge parameters were fitted using the HF/STO-3G method based on the antechamber program to construct GAFF force field. Owing to the lack of selenium and tellurium-related force field parameters in Amber force field, UFF force field was selected for molecular dynamics simulation of DTX-SSeS-DTX and DTX-STeS-DTX systems. Using a TIP3P water box model, the side length of the water box was set to 1 nm. First, 5000 steps of energy minimization were performed, followed by short-term 100 ps simulation under the NVT and NPT ensembles respectively. The cutoff radius was set to 1.0 nm. Configurations were maintained at 10 ps intervals for subsequent analysis. All ab initio calculations were performed based on Gaussian 09, and molecular dynamics simulations were accomplished utilizing Gromacs 2018. Hydrogen, hydrophobic, and chalcogen bonds were analyzed using Discovery Studio 4.5. In order to get closer to the real experiment, we further performed the molecular simulation adding DSPE-PEG$_{2K}$ and ethanol into the system according to the actual ratio. At the same time, the number of prodrug molecules increased to 80.

## Redox dual-responsive bioactivation

In release studies, PBS (pH 7.4) containing ethanol ($v/v = 30\%$) was used as the medium to meet the sink condition. Respectively, the HPNAs (100 nmol, 0.2 mL) were dispersed in 30 mL the medium with $H_2O_2$ (1 mM, 10 mM, 50 mM), DTT (0.01 mM, 0.1 mM, 1 mM) and GSH (0.1 mM), or without redox substance at $37\,°C$ with gentle shaking ($n = 3$). The concentration of the released DTX was measured using HPLC (Empower 3) with sampling (0.2 mL) at prespecified time intervals. To investigate the mechanism of redox-responsive drug release, HPNAs were incubated in 50 mM $H_2O_2$ or 50 mM DTT contained medium at $37\,°C$. Intermediates of the prodrug during release were confirmed by mass spectrometry (Waters Xevo TQ MS Detector, Waters, USA) using software MassLynx V4.1. The type of mass analyzer is the Triple Quadrupole. The ESI source was used for mass spectrometry and electrospray+ for ion mode. The MS1 was adopted as the acquisition mode. The capillary was 3.45 kV, the cone was 64 V, the collision was 0 and the desolvation gas flow was $650\,L\,h^{-1}$. In addition, we investigated the intracellular intermediates of HPNAs. After 6 h of drug treatment, the cells were broken by ultrasound, and the intermediates of prodrugs were extracted by acetonitrile. The molecular weight of the intermediate was determined by mass spectrometry (LCMS-8060, Shimadzu, Kyoto, Japan). LCMS-8060 is composed of a liquid phase system (LC-20AD XR) and a mass spectrometry system (LCMS-8060). The liquid mobile phase was acetonitrile and water ($v/v = 80:20$), both containing 0.1% formic acid, ran at a flow rate of $0.2\,mL\,min^{-1}$, and the sample volume was 5 µL. The type of mass analyzer is the Triple Quadrupole. MS acquisition settings adopted Acquisition Mode: Precursor Ion Scanning; Ion Mode: E+; Determination Time: 0.8 min; Event Time: 1 s; Collision Energy: 0.0 v; Nebulizer Flow: $3\,L\,min^{-1}$; Drying Gas Flow: $10\,L\,min^{-1}$. Software LabSolutions 6.72 was used to acquire mass spectrum of intermediates. Select integration algorithm: data processor, automatically according to the area of integration. Smoothing method was standard.

## Cellular redox heterogeneity

Intracellular ROS level was investigated by the DCF fluorescence using flow cytometer (Becton Dickinson, USA). 4T1 cells, B16F10 cells, Hepa 1–6 cells, or 3T3 cells were seeded on the cell climbing sheet and incubated in 12-well plates ($1 \times 10^5$ cells/well) for 24 h. The medium was replaced by fresh medium containing DCFH-DA (10 µM) for 30 min at $37\,°C$. The cells were washed three times with PBS and collected for analyzing by flow cytometer (Becton Dickinson, USA).

To detect intracellular GSH concentrations, cells were seeded into 6-well plates ($2 \times 10^5$ cells/well) and cultured. After for 24 h, the cells were washed with PBS, centrifuged and collected. Then the deproteinized reagent M solution was added. The glutathione assay kit (Beyotime, Shanghai, China) was used to determine intracellular GSH concentrations.

## Cytotoxicity assays and cellular uptake

The cytotoxicity of HPNAs were investigated by MTT assays. In 96-well plates, 4T1 cells, B16F10 cells, Hepa 1–6 cells, or 3T3 cells (1000 cells/well) were incubated for 24 h. After that, treated with various concentrations of HPNAs or DTX-solution for another 48 h ($n = 3$ for each concentration). After that, the cells were treated with 35 µL of MTT ($5\,mg\,mL^{-1}$) solution. After 4 h, the formazan crystal was dissolved by dimethyl sulfoxide. The absorbance was determined by an enzyme-labeled instrument (SYNERGY, BioTek Instruments, Inc, USA). The $IC_{50}$ values were calculated by GraphPad Prism 8, using molar concentration and cell viability ratio as parameter. The selectivity index (SI) was obtained by the ratio of $IC_{50}$ values of 3T3 cell line to tumor cells line.

To investigate the cellular uptake of HPNAs, 4T1 cells were incubated in 24-well plates ($5 \times 10^4$ cells per well) which cell slides were previously placed for 24 h. Subsequently, coumarin-6-labeled HPNAs (10 µg mL$^{-1}$, $w/w = 40:1$) or free coumarin-6 ($250\,ng\,mL^{-1}$) was added with into plates and incubated for 0.5 h or 2 h. After washing and fixing, the nucleus was counterstained by Hoechst 33342. The cell imaging was acquired by confocal laser scanning microscopy (CLSM, C2SI, Nikon, Japan) using software NIS 4.13. 4T1 cells were seeded into 12-well plates ($10^5$ cells per well) for 24 h with the above drug treatment scheme. The cells were collected and measured by flow cytometer (Becton Dickinson, USA).

## Apoptosis assay

4T1 cells were seeded into 12-well plates ($10^5$ cells per well) for 24 h. The 1640 medium containing Taxotere or HPNAs was added for 24 h (DTX equivalent of $200\,ng\,mL^{-1}$). Subsequently, the medium was collected and the cells were digested with trypsin without EDTA. Then the samples were treated according to the procedure suggested in the Annexin V-FITC/PI apoptosis detection kit (Solarbio, China). Apoptosis was measured by flow cytometer (Becton Dickinson, USA) using software BD CellQuest Pro and the results were analyzed using FlowJo software.

## In vitro microtubule polymerization assay

4T1 cells were grown in cell dishes and cultured for 24 h. Then, the culture medium was replaced with fresh medium containing Taxotere or HPNAs (100 nM equivalent to DTX), and further cultured for 48 h. After washing and fixing, the nucleus was counterstained by Hoechst 33342. Alexa Fluor 555 labeled anti-α-Tubulin mouse monoclonal antibody (clone name: DM1A) was diluted at 1:100. The cells were

treated according to the procedure suggested in Microtubule Tracker Red detection kit (Baiaolaibo technology co., ltd, China) and observed using confocal laser scanning microscopy (CLSM, C2SI, Nikon, Japan). Images were analyzed using Image J software.

## Effect on intracellular ROS and GSH level

The effect of HPNAs on intracellular ROS levels was investigated by the DCF fluorescence using CLSM and flow cytometer (Becton Dickinson, USA). 4T1 cells were seeded on the cell climbing sheet and incubated in 12-well plates ($10^5$ cells/well) for 24 h. This was followed by treatment with DTX-solution or HPNAs (100 μg mL$^{-1}$ equivalent to DTX) for 48 h. After that, the medium was replaced by fresh medium containing DCFH-DA (10 μM) for 30 min at 37 °C. Confocal microscopy was used to observe the cells (CLSM, C2SI, Nikon, Japan). For quantitative analysis, the intracellular ROS levels were also analyzed by flow cytometer. 4T1 cells were incubated in 12-well plates ($10^5$ cells/well) for 24 h. Then cells were treated with DTX-solution and HPNAs (200 nM equivalent to DTX) for 12 h. After that, the medium was replaced by fresh medium containing DCFH-DA (10 μM) for 30 min at 37 °C. The cells were washed three times with PBS, digested by trypsin, centrifuged, and resuspended by PBS. The results were analyzed using FlowJo software. To detect intracellular GSH concentrations, cells were seeded into 6-well plates ($2 \times 10^5$ cells/well). After 24 h, the cells were treated with DTX-solution and HPNAs (200 nM equivalent to DTX) for 12 h. Then, the cells were washed with PBS, centrifuged and collected. The deproteinized reagent M solution was added. The glutathione assay kit (Beyotime, Shanghai, China) was used to determine intracellular GSH concentrations.

## Intracellular bioactivation

To explore the intracellular drug release, 4T1 cells were incubated in 24-well plates for 24 h ($5 \times 10^4$ cells/well). Then treated with DTX-solution or HPNAs (50, 100, and 200 ng mL$^{-1}$, equivalent to DTX) for 48 h and 72 h. Three samples in parallel for each concentration. Then, the cells and the culture medium were collected. The cells were disrupted by ultrasound. The concentrations of DTX were measured by UPLC-MS-MS system (Xevo TQ, Waters, USA) using software MassLynx V4.1. Xevo TQ (Waters, USA) is composed of a liquid phase system (ACQUITY Binary Solvent Manager Instrument) and a mass spectrometry system (Waters Xevo TQ MS Detector). The type of mass analyzer is the Triple Quadrupole. The liquid mobile phase was acetonitrile and water ($v/v = 80: 20$), both containing 0.1% formic acid, ran at a flow rate of 0.2 mL min$^{-1}$, and the sample volume was 5 μL. We used a reversed-phase chromatography column (Kinetex 2.6 μm XB-C18 50*2.1 mm) at the column temperature of 30 °C. The ESI source was used for mass spectrometry and electrospray+ for ion mode. The MSMS was adopted as the acquisition mode. The capillary was 3.45 kV, the cone was 64 V, the collision was 64, and the desolvation gas flow was 650 L h$^{-1}$.

## Pharmacokinetic study of HPNAs

HPNAs were added to rat plasma in a ratio of 1:9 ($v/v$) to evaluate chemical stability. Specifically, 50 μL HPNAs (1 mg mL$^{-1}$) was added to 450 μL fresh rat plasma and mixed evenly by the vortex. At the predetermined time, a 50 μL sample was collected, added to 150 μL acetonitrile, and vortexed for 5 min. The acetonitrile could precipitate plasma proteins and dissolve the prodrugs and DTX. Moreover, acetonitrile was used as a demulsifier to extract prodrugs from HPNAs. The samples were centrifuged at $11,000 \times g$ for 10 min to separate the precipitated protein from acetonitrile. The content of the prodrug in acetonitrile was determined by HPLC with the external standard method, using a 25 cm COSMOSIL® 5C18-PAQ column, acetonitrile/water as mobile phase ($v/v = 80:20$). To measure the pharmacokinetic profiles of HPNAs, Sprague-Dawley rats (male, 6 weeks old) were randomly grouped and intravenously administered Taxotere or HPNAs at

a dose of 4 mg kg$^{-1}$ (equivalent to DTX, $n = 3$). The blood was taken from the rat's posterior ocular venous plexus and plasma was isolated at 2 min, 5 min, 15 min, 30 min, 1 h, 2 h, 4 h, 8 h, 12 h, and 24 h, respectively. The plasma concentration of prodrugs and free DTX was measured via UPLC-MS-MS system (Xevo TQ, Waters, USA) using software MassLynx V4.1. Xevo TQ (Waters, USA) is composed of a liquid phase system (ACQUITY Binary Solvent Manager Instrument) and a mass spectrometry system (Waters Xevo TQ MS Detector). The type of mass analyzer is the Triple Quadrupole. Because of the low concentration of drugs in plasma, we used liquid-liquid extraction to recover prodrugs from plasma samples. Specifically, a 200 μL plasma sample was collected, added with 200 μL of acetonitrile and 100 μL of internal standard (PTX, 500 ng mL$^{-1}$), and then vortexed for 3 min. After that, 3 mL methyl tert-butyl ether was added to the sample, vortexed for 3 min, and centrifuged at $1800 \times g$ for 5 min. The solvent was volatilized under nitrogen, and 100 μL acetonitrile was added to redissolve the sample. The supernatant was collected after vortex and centrifugation. The liquid mobile phase was acetonitrile and water ($v/v = 80:20$), both containing 0.1% formic acid, ran at a flow rate of 0.2 mL min$^{-1}$, and the sample volume was 5 μL. We used a reversed-phase chromatography column (Kinetex 2.6 μm XB-C18 50*2.1 mm) at the column temperature of 30 °C. The ESI source was used for mass spectrometry and electrospray+ for ion mode. The MSMS was adopted as the acquisition mode. The capillary was 3.45 kV, the cone was 64 V, the collision was 64, and the desolvation gas flow was 650 L h$^{-1}$. The parent ion of the DTX-STeS-DTX prodrug was 1941.5, daughter ion was 889, and the predicted retention time was 1.135 min. The parent ion of the DTX-SSeS-DTX prodrug was 1892.5, daughter ion was 839, and the predicted retention time was 1.035 min. The parent ion of the DTX-SSS-DTX prodrug was 1844.5, daughter ion was 791.1, and the predicted retention time was 0.960 min. The parent ion of the DTX-SCS-DTX prodrug was 1826.5, daughter ion was 773.3, and the predicted retention time was 0.920 min. DAS 2.1.1 software was used to calculate the pharmacokinetic parameters. Pharmacokinetic Calculations-Batch Data Analysis-Statistical Moment Parameters were used for processing.

## Biodistribution of HPNAs

To investigate the biodistribution of HPNAs, 4T1 tumor models were established on BALB/c mice (female, 11 weeks old). Mice were randomly grouped. DiR-labeled HPNAs and DiR-solution were intravenously administrated at a dose of 0.5 mg kg$^{-1}$ equivalent to DiR until tumor volume reached around 300 mm$^3$ ($n = 3$). The fluorescence signal was acquired utilizing an in vivo imaging system (IVIS) spectrum after 4 h and 12 h post-administration. To evaluate the DTX release in tumors of HPNAs, the 4T1 tumor model was established. The mice were treated with 20 mg kg$^{-1}$ Taxotere or HPNAs ($n = 3$). After 1 h, 12 h, and 24 h post-injection, the concentrations of DTX in the tumor were determined by UPLC-MS-MS (Waters Co., Ltd., Milford, MA, USA). Specifically, 1 mL acetonitrile was added to 500 mg tumor tissue, and the sample was sheared using a homogenizer. Then, the supernatant was collected after centrifugation ($1800 \times g$, 10 min). 100 μL internal standard was added in 500 μL supernatant and vortexed for 3 min. The supernatant was collected after centrifugation again ($11,000 \times g$, 5 min). The solvent was volatilized under nitrogen, and 100 μL acetonitrile was added to redissolve the sample. The supernatant was collected after vortex and centrifugation. Liquid mobile phase was acetonitrile and water ($v/v = 80:20$), both containing 0.1% formic acid, using a reversed-phase chromatography column (Kinetex 2.6 μm XB-C18 50*2.1 mm). The ESI source was used for mass spectrometry and electrospray+ for ion mode. The capillary was 3.45 kV, the cone was 64 V, and the collision was 64.

## Antitumor efficacy of HPNAs

The 4T1 tumor-bearing mice model was established by subcutaneously implanted of 4T1 cells ($5 \times 10^6$) into the right-back of female BALB/c

mice (female, 11 weeks old). At the tumor volume reached ~100 mm³, mice were randomly grouped and intravenously administered the Taxotere, HPNAs, or saline at a dose of 2 mg kg⁻¹ (equivalent to DTX, $n = 5$). This day was designated as day 1 for administration. Preparations were administered via intravenous injection every other day for a total of five injections. Where mice exceeded 20% loss in body weight, drug administration was stopped, and mice were given nutrition substances and closely monitored. This was approved by the Institutional Animal Care and Use Committee (IACUC) of Shenyang Pharmaceutical University. The tumor volume was calculated as follows: Tumor volume (mm³) = (length × width × width)/2. On day 10, the mice were sacrificed, and the tumors were isolated and weighed. The lung metastatic nodules were observed after soaking and staining with Bouin's tissue fixative solution. In addition, the apoptosis and proliferation of the tumor sections were investigated by TUNEL and Ki-67 staining. Ki67 Rabbit polyclonal antibody was used with dilution (1: 1000).

In addition, in vivo antitumor effect at an intermediate dose (10 mg kg⁻¹) of four HPNAs and high dose (15 mg kg⁻¹ and 20 mg kg⁻¹) of DTX-STeS-DTX NPs were investigated on BALB/c mice (female, 11 weeks old) bearing 4T1 tumor. The implantation of tumor and the treatments were carried out using the same experimental schemes.

### Safety
Body weights were monitored during the above pharmacodynamics. After the pharmacodynamics, the whole blood was collected for blood analysis. The serum was obtained for hepatorenal function analysis. The main organs (heart, liver, spleen, lung, and kidney) and tumor tissues were collected for hematoxylin and eosin (H&E) staining.

### Statistical analysis
Data were calculated as mean value ± standard deviation. Statistical differences between groups were analyzed with Student's $t$-test (two-tailed) and one-way ANOVA. The exact $P$-value of is provided in the corresponding figure. Statistical significance was considered at n.s (no significance) $P > 0.05$, *$P < 0.05$, **$P < 0.01$, ***$P < 0.001$, and ****$P < 0.0001$.

### Reporting summary
Further information on research design is available in the Nature Portfolio Reporting Summary linked to this article.

## Data availability
All data supporting the findings of this study are available within the Article, Supplementary Information or Source Data file. The source data underlying Fig. 2c–e, Fig. 3c–d, Fig. 4a–f, Fig. 5a–d, f, h, Fig. 6a–c, e–g, Fig. 7a–c, e–g, i–k, supplementary Figs. 9, 12, 15, 16, 28, 32, 33, 35, 36, 37, 40, 41, 42, 43, 45, 46, 47, 49, and 50 have been deposited in the Figshare database (https://doi.org/10.6084/m9.figshare.20941144)[49]. Source Data.xlsx (figshare.com) Source data are provided with this paper.

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

## Acknowledgements

This work was financially supported by National Key R&D Program of China (No. 2022YFE0111600, J.S.), National Natural Science Foundation of China (No. 82073777, J.S.), National Natural Science Foundation of China (No. 81872816, Z.H.), Doctoral Scientific Research Staring Foundation of Liaoning Province (NO. 2021-BS–130, B.S.), General Program of Department of Education of Liaoning Province (NO. LJKZ0953, B.S.).

## Author contributions

All authors have given approval to the final version of the manuscript. J.S., B.S., and T.L. conceived the project. B.S and T.L. designed experiments; T.L. and B.S. performed research; T.L., L.L., S.W., F.D., S.Z., J.S., X.W., Q.L., and H.W. executed the experiments and analyzed data; H.Z., M.C., X.L., and Z.H. provided useful suggestions; T.L., B.S., and J.S. wrote the paper.

## Competing interests

The authors declare no competing interests.
