## [Peer Review File · Nature Communications]

Reviewers' Comments:

Reviewer #1:

Remarks to the Author:

This manuscript follows on from some prior work by the authors on chalcogen-link polymer prodrugs and the use of Se and Te in redox-based release systems. There are extensive data in the manuscript, the chalcogen-link DTX dimers are very well-characterised and there is convincing evidence of their identity and purity. The DTX dimers were then co-precipitated with PEG-DSPE and dosed in mice with 4T1 tumors grown in the flanks of Balb C mice. The greatest efficacy was shown with DTX-S-Te-S-DTX prodrugs at a dose of 15 mg/kg and 20 mg/kg, with good tolerability in terms of mouse body weight compared to taxotere controls. The formulations are certainly new and there seem to be some possibilities for this technology, but the manuscript contains some major omissions which need to be addressed and some flaws in terms of scientific logic. There are also issues regarding whether the work could be repeated by others and an unacceptable lack of ethics statements. Publication is not recommended unless these questions can be fully resolved.

1. There is a logical disconnect in relating the computational structures with those prepared in the lab. In the computations, the self-assembly of the HPNAs took place in a 'box' of water, with no ions or solvent present. Accordingly, the interactions are only those of the HPNAs with themselves and any solvophobic interactions, as H-bonding would not be significant for the DTX moieties. In the lab, preparation of the PEG-DSPE-coated DTX homodimeric HPNAs was carried out by taking mixtures of the dimers and the PEG-surfactants dissolved in ethanol and adding to water to cause precipitation. In this 'real-world' process, ethanol molecules diffuse away into bulk water, while the HPNAs co-precipitate with the PEG-DSPE, with the PEG chains providing some level of colloidal stabilisation. It is thus very tenuous to imply that self-assembly in the PEG-DSPE-co-HPNA particles in ethanol/water follows the predicted pattern for HPNAs alone. In addition, if one looks at the sizes of the HPNAs, the end-to-end distance is ~ 3 nm yet the particles are ~ 30 times bigger than this, so must contain multiple assemblies kinetically trapped alongside the PEG-DSPE. For the self-assembled HPNAs without surfactant the sizes are at least 200 nm, so must involve even more molecular aggregation. How then do the authors account for self-assembly of multiple HPNAs being related to the S-X-S bond system?

2. It would not be possible for others to reproduce the experiments relating to biodistribution. For example "HPNAs were added to rat plasma in a ratio of 1: 9 (v/v) to evaluate chemical stability. At predetermined time points, 50 μ L samples were precipitated protein and determined by HPLC" No actual volumes are given, and the meaning of 'precipitated protein' is not clear. How were the DTX dimer concentrations and amounts actually measured? What LC-MS conditions, columns etc were used? How were the prodrugs and DTX separated from the NPs for the analysis?

3. Similar questions apply to the pharmacokinetic profiles experiments, i.e. how were the drugs/pro-drugs recovered from the NPs prior to LC-MS.?

4. The labelling with DiR is also not described - how can the authors be sure that what they observe is not dye which leached out of the NPs?

Minor errors

There is excessive self-citation – the broader context for this work is not given.

The statement "BALB/c mice. DiR labeled HPNAs showed much higher fluorescent signals in tumors than Di R solution (Fig. 6 D F)). These results proved that HPNAs could effectively accumulate in tumors by EPR effects" should be revised. The data indicate that labelled nanoparticles or free DiR were detected in tumors but no evidence for EPR was provided.

Error bars in Figure S9, S19. Only for some of the curves in these figures are error bars visible.

Either they need to be added, or the scales need to be adjusted, or split axes used, so that the errors can be seen.

Supplementary Table 1 – quote sizes only to 2 decimal places, i.e. 96 nm not 96.12 nm as DLS not accurate to this level.

Reviewer #2:

Remarks to the Author:

The authors reported that the introduction of hybrid chalcogen bonds enhanced the stability, and bioactivation of docetaxel dimeric prodrug nanoassemblies in tumor tissues. The STeS hybrid bond exhibited redox dual-responsivity, and the response mechanism is interesting. Overall, the work is well designed with novelty. However, it needs improvement as follows.

Comments:

1. The authors investigated the drug release in vitro and detected the intermediates. However, the intracellular environment is more complicated than that in vitro. The authors need to detect the intracellular intermediates of prodrugs to verify the release mechanism.
2. The chemical structure of DTX in Figure 2 is suggested to be omitted, and thus the different bridged structures could be shown clearly.
3. Generally, the reductive bridged bond is responsive to GSH under physiological conditions. The authors should investigate the drug release of the prodrug nanoparticles in the solutions with GSH rather than DTT.
4. In Figure 4a-f, the drug release profile should be conducted in the condition without DTT and H₂O₂ as control.
5. In vitro drug release, the mixed solution of PBS and ethanol was chosen as the release medium. The authors should explain the rationality of the choice of release medium.
6. In Figure 5A-D, the "DTX" used by the author is confusing. Does it represent DTX solution or Taxotere?
7. The cellular uptake results should be quantified by flow cytometry.
8. In Figure 5, inhibition of tumor cells and normal cell by prodrug nanoassemblies was examined. Meanwhile, the difference in the redox state of cells needs to be investigated.
9. The microtubule polymerization assay is recommended to verify cytotoxicity.
10. The discussion part of the manuscript needs to be improved. It is suggested that the author objectively analyze the limitations of the research and the problems that need to be solved in the process of clinical transformation.

Reviewer #3:

Remarks to the Author:

In this manuscript, the hybrid chalcogen bond was creatively introduced to the prodrug nanoassemblies, and its effect on the self-assembly, redox-responsivity and antitumor efficacy of DTX dimeric prodrugs was investigated in comparison with the previous trisulfide bond. The authors evaluated the self-assembly ability, bioactivation, pharmacokinetic behavior, biodistribution, and pharmacodynamics, and gained in-depth insight into the advantages of hybrid chalcogen bond on the dimeric prodrug nanoassemblies to address heterogeneous tumor redox-microenvironment. Overall, this study is innovative and the data is interesting. However, the following aspects should be addressed.

1. The prodrug nanoassemblies have been further loaded into DSPE-PEG2K for in vivo application, and this loading strategy may compromise "the high drug loading" advantage of dimer prodrug. Could the author give more explanation for the rationale of this design?
2. The storage stability of nanoassemblies for a longer period of time needs to be investigated.
3. In Figure 4, compared with reduction conditions, prodrug nanoassemblies need more oxidative stimuli to release drugs, which the authors need to explain. In addition, the degradation of prodrugs should be investigated to prove the oxidative response of hybrid bonds.
4. The heterogeneity of redox levels in different cell lines needs to be investigated. The authors should supplement the data of ROS and GSH content in different cells.
5. Selenium and tellurium, as microelements in human body, are involved in the regulation of redox balance. Therefore, the author needs to consider whether the hybrid chalcogen bond containing prodrug nanoassemblies will affect the redox level of tumor cells.
6. Details need to be provided about how IC₅₀ values and tumor volume were calculated.
7. The writing needs further polishing. I suggest that the author standardize the use of technical term to avoid misunderstanding to readers. For example, whether "Mean Diameter" and "Size" mean the same meaning.
8. The author should pay attention to some details, such as the missing space between ordinate

name and unit in Figure 2E.

Manuscript ID: NCOMMS-22-08741A

Title: Hybrid Chalcogen Bond as the “Double-Control Switch” of Homodimeric Prodrug Nanoassemblies to Address Tumor Redox-Heterogeneity

Dear reviewers,

We are truly grateful to your valuable comments and hard work. Based on the comments and suggestions, we have made further modifications on the original manuscript. The corrected portions in the revised manuscript have been highlighted in red. Below we summarize a point-by-point response to each of the comments from the reviewers.

We appreciate the critical reviews of the manuscript and hope the revised version attached is improved to a significant degree to merit further consideration of publication.

Response to Reviewers' comments:

Reviewer 1:

This manuscript follows on from some prior work by the authors on chalcogen-link polymer pro-drugs and the use of Se and Te in redox-based release systems. There are extensive data in the manuscript, the chalcogen-link DTX dimers are very well-characterised and there is convincing evidence of their identity and purity. The DTX dimers were then co-precipitated with PEG-DSPE and dosed in mice with 4T1 tumors grown in the flanks of Balb C mice. The greatest efficacy was shown with DTX-S-Te-S-DTX prodrugs at a dose of 15 mg/kg and 20 mg/kg, with good tolerability in terms of mouse body weight compared to taxotere controls. The formulations are certainly new and there seem to be some possibilities for this technology, but the manuscript contains some major omissions which need to be addressed and some flaws in terms of scientific logic. There are also issues regarding whether the work could be repeated by others and an unacceptable lack of ethics statements. Publication is not recommended unless these questions can be fully resolved.

Response: We appreciate the reviewer's comments. According to your suggestion, we have revised the omissions and defects of scientific logic in the manuscript. The details of the experimental method were supplemented so that readers can reproduce our results. In addition, the explanation of animal ethics was added in the manuscript. The specific answers to your comments were replied one by one below.

1. There is a logical disconnect in relating the computational structures with those prepared in the lab. In the computations, the self-assembly of the HPNAs took place in a 'box' of water, with no ions or solvent present. Accordingly, the interactions are only those of the HPNAs with themselves and any solvophobic interactions, as H-bonding would not be significant for the DTX moieties. In the lab, preparation of the PEG-DSPE-coated DTX homodimeric HPNAs was carried out by taking mixtures of the dimers and the PEG-surfactants dissolved in ethanol and adding to water to cause precipitation. In this 'real-world' process, ethanol molecules diffuse away into bulk water, while the HPNAs co-precipitate with the PEG-DSPE, with the PEG chains providing some level of colloidal stabilisation. It is thus very tenuous to imply that self-assembly in the PEG-DSPE-co-HPNA particles in ethanol/water follows the predicted pattern for HPNAs alone. In addition, if one looks at the sizes of the HPNAs, the end-to-end distance is ~ 3 nm yet the particles are ~ 30 times bigger than this, so must contain multiple assemblies kinetically trapped alongside the PEG-DSPE. For the self-assembled HPNAs without surfactant the sizes are at least 200 nm, so must involve even more molecular aggregation. How then do the authors account for self-assembly of multiple HPNAs being related to the S-X-S bond system?

Response: We appreciate the reviewer's comments. First, we understand the doubts that the previous molecular dynamics simulation did not reflect the preparation process. In the previous manuscript, we focused on the influence of structural difference on the assembly ability, so DSPE-PEG_{2K} and ethanol were not added. The water we used in the experiment was deionized, so no ions were added to the initial simulation system. In addition, the particle size of the simulated assembly was smaller than the actual

nanoparticles since we only used 20 prodrug molecules for dynamic microsimulation in previous manuscripts. Too many molecules require a more extensive system and calculation, and the current computer can't calculate such a large system as the “real world”. To get closer to the natural experiment, we re-performed the molecular simulation based on the reviewer’s suggestion, adding ethanol molecules and DSPE-PEG_{2K} into the initial system according to the actual ratio. At the same time, the number of prodrug molecules was increased by four times (n=80). The calculation results in the new system showed that the assembly ability of prodrugs was consistent with the previous system.

In the self-assembly process, most of the aggregated force was the hydrophobic force provided by DTX, and S-X-S provides the key steric hindrance for the assembly formation to prevent excessive aggregation and precipitation of molecules. Because the structural difference of the four prodrugs is concentrated on the linkers (S-X-S bond), we inferred that the difference in linkers caused the difference in assembly.

The results were discussed in the revised manuscript (page 10): “Furthermore, according to the preparation process of HPNAs, we added ethanol and DSPE-PEG_{2K} in a corresponding proportion to the dynamic simulation system. In addition, the number of prodrug molecules was increased by up four times (n=80) closer to the studied experiment (Supplementary Fig. 10). Similar to the above results, DTX-STeS-DTX prodrugs were assembled fastest and had the lowest binding energy, followed by DTX-SSeS-DTX prodrugs, DTX-SSS-DTX prodrugs, and finally DTX-SCS-DTX prodrugs (Supplementary Fig. 11 and Supplementary Table 2). The radiuses of gyration of the four systems were shown in Supplementary Fig. 12, and the time length to reach equilibrium was DTX-STeS-DTX < DTX-SSeS-DTX < DTX-SSS-DTX < DTX-SCS-DTX. These results further suggested the best assembly efficiency of DTX-STeS-DTX”.

Supplementary Figure 10. The initial conformation of the system. (The red represents water, the gray structure represents ethanol, the blue structure represents DSPE-PEG_{2K}, and the green structure represents prodrugs.)

Supplementary Figure 11. Track of self-assembly conformation of prodrug in the system with time. (The blue structure represents DSPE-PEG_{2K}, and the green structure represents prodrugs.)

Supplementary Figure 12. The radius of gyration of prodrug and DSPE-PEG_{2K} during aggregation.

Supplementary Table 2. Total binding energy of prodrug and DSPE-PEG_{2K} system.

Contribution (kJ/mol)	DTX-SCS- DTX	DTX-SSS- DTX	DTX-SSeS- DTX	DTX-SteS- DTX
ΔE_{vdW}	-13032.8	-14887.9	-15736.9	-16832.9
ΔE_{elec}	-1123.1	-1205.2	-1324.7	-1407.8
ΔG_{polar}	5251.0	5846.6	5364.5	5443.9
$\Delta G_{nonpolar}$	-1071.2	-1225.7	-1351.7	-1458.9
ΔG_{total}	-9976.1	-11472.2	-13048.8	-14255.7

Total binding energy (ΔG_{total}) = Van der Waals (ΔE_{vdW}) + Electrostatic energy (ΔE_{elec}) + Polarization solvation energy (ΔG_{polar}) + Nonpolarized solvation energy ($\Delta G_{nonpolar}$)

2. It would not be possible for others to reproduce the experiments relating to biodistribution. For example “HPNAs were added to rat plasma in a ratio of 1: 9

(v/v) to evaluate chemical stability. At predetermined time points, 50 µL samples were precipitated protein and determined by HPLC” No actual volumes are given, and the meaning of ‘precipitated protein’ is not clear. How were the DTX dimer concentrations and amounts actually measured? What LC-MS conditions, columns etc were used? How were the prodrugs and DTX separated from the NPs for the analysis?

Response: We appreciate the reviewer’s comments. The supplement of these experimental methods will make the article more complete and facilitate readers to reproduce the experiment. In the section of method, we have described the specific steps of the experimental process in detail. In the plasma chemical stability of HPNAs (page 36), “Specifically, 50 µL HPNAs (1mg/mL) was added to 450 µL fresh rat plasma and mixed evenly by the vortex. At the predetermined time, a 50 µL sample was collected, added to 150 µL acetonitrile, and vortexed for 5 min. The acetonitrile could precipitate plasma proteins and dissolve the prodrugs and DTX. Moreover, acetonitrile was used as a demulsifier to extract prodrugs from HPNAs. The samples were centrifuged at 13000 rpm for 10 min to separate the precipitated protein from acetonitrile. The content of the prodrug in acetonitrile was determined by HPLC with the external standard method, using a 25 cm COSMOSIL® 5C18-PAQ column, acetonitrile/water as mobile phase (v/v = 80: 20)”. In this study, acetonitrile could precipitate proteins and dissolve drugs in plasma. More importantly, it was used as a demulsifier to destroy the structure of HPNAs, thus dissolving and extracting the drugs in HPNAs. Here, we aim to investigate the chemical stability of HPNAs in rat plasma, so we did not separate the dissolved prodrugs and DTX from the HPNAs in plasma. All prodrugs and DTX (including the part disassembled in plasma and the prodrugs in HPNAs) were dissolved and extracted with acetonitrile for the analysis.

In addition, we have refined and improved the methods of other experiments. In the tumor bioactivation experiment (page 37): “Specifically, 1 mL acetonitrile was added to 500 mg tumor tissue, and the sample was sheared using a homogenizer. Then, the supernatant was collected after centrifugation (4000 rpm, 10 min). 100 µL internal standard was added in 500 µL supernatant and vortexed for 3 min. The supernatant was

collected after centrifugation again (13000 rpm, 5 min). The solvent was volatilized under nitrogen, and 100 μ L acetonitrile was added to redissolve the sample. The supernatant was collected after vortex and centrifugation. Liquid mobile phase was acetonitrile and water (v/v=80: 20), both containing 0.1% formic acid, using a reversed-phase chromatography column (Kinetex 2.6 μ m XB-C18 50*2.1 mm). The ESI source was used for mass spectrometry and electrospray+ for ion mode. The capillary was 3.45 kV, the cone was 64 V, and the collision was 64”.

3. Similar questions apply to the pharmacokinetic profiles experiments, i.e. how were the drugs/pro-drugs recovered from the NPs prior to LC-MS.?

Response: We supplemented the method for recovering drugs/prodrugs from the NPs in the revised manuscript (page 36): “The blood was taken from the rat’s posterior ocular venous plexus and plasma was isolated at 2 min, 5 min, 15 min, 30 min, 1 h, 2 h, 4 h, 8 h, 12 h, and 24 h, respectively. The plasma concentration of prodrugs and free DTX was measured via UPLC-MS-MS system (Waters Co., Ltd., Milford, MA, USA). Because of the low concentration of drugs in plasma, we used liquid-liquid extraction to recover prodrugs from plasma samples. Specifically, a 200 μ L plasma sample was collected, added with 200 μ L of acetonitrile and 100 μ L of internal standard (PTX, 500 ng/mL), and then vortexed for 3 min. After that, 3 mL methyl tert-butyl ether was added to the sample, vortexed for 3 min, and centrifuged at 4000 rpm for 5 min. The solvent was volatilized under nitrogen, and 100 μ L acetonitrile was added to redissolve the sample. The supernatant was collected after vortex and centrifugation. The liquid mobile phase was acetonitrile and water (v/v=80: 20), both containing 0.1% formic acid, using a reversed-phase chromatography column (Kinetex 2.6 μ m XB-C18 50*2.1 mm). The ESI source was used for mass spectrometry and electrospray+ for ion mode. The capillary was 3.45 kV, the cone was 64 V, and the collision was 64”. As mentioned before, we did not separate the dissolved prodrugs and DTX from the HPNAs. All prodrugs, DTX, and HPNAs were dissolved and extracted by acetonitrile.

4. The labelling with DiR is also not described - how can the authors be sure that

what they observe is not dye which leached out of the NPs?

Response: We appreciate the reviewer's comments. DiR is widely used to label nanoparticles and investigate their tissue distribution. (*Nat Nanotechnol.* 2022, 17, 206-216; *Nat Nanotechnol.* 2017, 12, 692-700; *Nat Biomed Eng.* 2021, 5, 983-997; *J Extracell Vesicles.* 2021, 10, 10, e12134.) However, a recognized problem is that the free DiR cannot be effectively distinguished from the DiR-labeled nanoparticles.

The distribution behavior of the DiR solution was inconsistent with that of HPNAs. The free DiR solution was hardly distributed in tumors, while DiR-labeled HPNAs were mainly distributed in the liver, spleen, and tumors. If DiR leaked during the systemic circulation from HPNAs, its biodistribution would be similar to the DiR solution. Therefore, the DiR fluorescence signal in the tumor was mainly composed of DiR-labeled HPNAs and DiR released from HPNAs in the tumor. In addition, we measured the release of DTX in tumor by more accurate liquid chromatography-mass spectrometry to verify the tumor accumulation of HPNAs (Fig. 6 G). At 24 h, DTX-STeS-DTX NPs exhibited significantly higher DTX concentration in tumors than the other groups.

Fig. 6. Pharmacokinetics and biodistribution of HPNAs. Pharmacokinetic profiles of the (A) prodrugs, (B) released DTX, and (C) total equivalent DTX. (D) Biodistribution of DiR-solution or DiR-labeled HPNAs. Quantitative analysis of biodistribution at (E) 4 h and (F) 12 h. (G) The accumulated DTX of Taxotere and HPNAs in tumor. Data were presented as mean \pm SD (n = 3). * $P < 0.05$, ** $P < 0.01$, and *** $P < 0.001$ by two-tailed Student's t-test.

Minor errors

1. There is excessive self-citation – the broader context for this work is not given.

Response: We appreciate the reviewer's comments. The related references of dimeric prodrugs and selenium-containing polymers were added to support the background. In the field of dimeric prodrugs, Pro. Chen Jiang and Pro. Zhigang Xie have made meaningful explorations. Professor Huaping Xu reported many studies on polymers

containing selenium or tellurium. Their relevant articles were added in the introduction: *J Nanobiotechnology*. 2021, 1, 19, 441; *Theranostics*. 2018, 18, 8, 4884-4897; *ACS Appl Mater Interfaces*. 2018, 46, 10, 39455-39467; *Bioorg Med Chem Lett*. 2017, 11, 27, 2493-2496; *Biomater Sci*. 2017, 8, 5, 1517-1521; *ACS Appl Mater Interfaces*. 2017, 32, 9, 26740-26748; *J Am Chem Soc*. 2015, 10, 137, 3458-3461; *Nano Today*. 2015, 6, 10, 717-736. In addition, some self-cited articles were removed from the manuscript.

2.The statement “BALB/c mice. DiR labeled HPNAs showed much higher fluorescent signals in tumors than DiR solution (Fig. 6 D F). These results proved that HPNAs could effectively accumulate in tumors by EPR effects” should be revised. The data indicate that labelled nanoparticles or free DiR were detected in tumors but no evidence for EPR was provided.

Response: We agree with the reviewer’s comments. We haven’t proved the EPR effect, so we deleted this part from the text. The results were corrected in the revised manuscript (page 21): “**These results proved that HPNAs could effectively accumulate in tumors.**”.

3.Error bars in Figure S9, S19. Only for some of the curves in these figures are error bars visible. Either they need to be added, or the scales need to be adjusted, or split axes used, so that the errors can be seen.

Response: We appreciate the reviewer’s comments. In GraphPad Prism software, the error bar of data was covered because the large size of the symbol . In the revised manuscript, we reduced the size of symbol, so that the error bar can be displayed.

Supplementary Figure 9. Non-PEGylated HPNAs stored at room temperature for 240

h.

Supplementary Figure 32. Cell viability treated with various concentrations of Taxol and HPNAs: (A) 4T1 cells, (B) B16-F10 cells, (C) Hepa 1-6 cells, and (D) 3T3 cells (n=3).

4. Supplementary Table 1 – quote sizes only to 2 decimal places, i.e. 96 nm not 96.12 nm as DLS not accurate to this level.

Response: Thanks to the reviewer's suggestion, we have revised the table.

Supplementary Table 1. Characterization of PEGylated HPNAs (n=3).

Nanoassemblies	Size (nm)	PDI	Zeta (mV)	DL (w/w, %)
DTX-STeS-DTX NPs	96	0.15	-24.3	67.38
DTX-SSeS-DTX NPs	95	0.19	-21.0	69.18
DTX-SSS-DTX NPs	97	0.14	-19.8	71.00
DTX-SCS-DTX NPs	109	0.12	-20.3	71.70

Reviewer 2:

The authors reported that the introduction of hybrid chalcogen bonds enhanced the stability, and bioactivation of docetaxel dimeric prodrug nanoassemblies in tumor tissues. The STeS hybrid bond exhibited redox dual-responsivity, and the response mechanism is interesting. Overall, the work is well designed with novelty. However, it needs improvement as follows.

Comments:

1. The authors investigated the drug release in vitro and detected the intermediates. However, the intracellular environment is more complicated than that in vitro. The authors need to detect the intracellular intermediates of prodrugs to verify the release mechanism.

Response: We appreciate the reviewer's comments. We investigated the intracellular intermediates of HPNAs. After 6 h of drug treatment, the cells were broken by ultrasound, and the intermediates of prodrugs were extracted by acetonitrile. The molecular weight of the intermediate was determined by mass spectrometry. As shown in Supplementary Fig. 21-24, 29-31, intracellular drug intermediates, including oxidation and reduction intermediates, were certified by mass spectrometry, and the

release mechanism was agreement with that in vitro.

Among them, oxidation-intermediates included a series of oxidation products such as sulfoxide and sulfone (Fig. 21-24). In addition, telluric acid, tellurite acid, selenite acid, and selenite acid were detected in cells treated with DTX-STeS-DTX NPs or DTX-SSeS-DTX NPs (Fig. 21-22). These results further proved that the release process of oxidative response existed in cells.

The reductive intermediates of prodrugs mainly include GSH-related intermediates DTX-SH, DTX-SXS-GSH, and GSH-SXS-GSH, while intermediates DTX-SXH and DTX-SS-GSH were not detected. These results verified that sulfhydryl substances preferentially attack the X atom of SXS rather than the adjacent S atom. In addition, intracellular reducing small-molecule such as cysteine were also involved in the reduction-responsivity of prodrugs. The corresponding intermediates of DTX-SXS-Cys and Cys-SXS-Cys were detected. The results were described in the revised manuscript (pages 13-15): “Moreover, we further investigated the intracellular intermediates of prodrugs to verify the release mechanism. The results showed that the intracellular oxidation intermediates of prodrugs were consistent with those in vitro, such as sulfoxide and sulfone (Supplementary Fig. 21-24). In addition, oxidative metabolites such as telluric acid, tellurite acid, selenite acid, and selenite acid were detected in cells treated with DTX-STeS-DTX NPs or DTX-SSeS-DTX NPs (Supplementary Fig. 21-22). These results further proved that the release process of oxidative response existed in cells”.

“Similarly, we verified the reduction intermediates in cells. We found that intracellular reducing substances, such as glutathione and cysteine, were involved in the reduction-responsivity of prodrugs. The intracellular reduction intermediates were shown in Supplementary Fig. 29-31, including DTX-SH, DTX-SXS-GSH, GSH-SXS-GSH, DTX-SXS-Cys, and Cys-SXS-Cys. This was the first time that we have verified the redox-response release mechanism of HPNAs at the cellular level”.

Supplementary Figure 21. Mass spectra of intracellular intermediates of DTX-STeS-DTX NPs. (A-E) monoxide to pentoxide of DTX-STeS-DTX. (F-G) tellurite acid and telluric acid.

Supplementary Figure 22. Mass spectra of intracellular intermediates of DTX-SSeS-DTX NPs. (A-F) monoxide to hexaoxide of DTX-SSeS-DTX. (G-H) selenous acid and selenic acid.

selenic acid.

Supplementary Figure 23. Mass spectra of intracellular intermediates of DTX-SSS-DTX NPs. (A-F) monoxide to hexaoxide of DTX-SSS-DTX.

Supplementary Figure 24. Mass spectra of intracellular intermediates of DTX-SCS-DTX NPs. (A-D) monoxide to tetroxide of DTX-SCS-DTX.

Supplementary Figure 29. Mass spectra of intracellular intermediates of DTX-STeS-DTX NPs. (A) DTX-SH. (B) DTX-STeS-GSH. (C) GSH-STeS-GSH. (D) DTX-STeS-Cys. (E) Cys-STeS-Cys.

Supplementary Figure 30. Mass spectra of intracellular intermediates of DTX-SSeS-DTX NPs. (A) DTX-SH. (B) DTX-SSeS-GSH. (C) GSH-SSeS-GSH. (D) DTX-SSeS-Cys. (E) Cys-SSeS-Cys.

Supplementary Figure 31. Mass spectra of intracellular intermediates of DTX-SSS-DTX NPs. **(A)** DTX-SH. **(B)** DTX-SSS-GSH. **(C)** GSH-SSS-GSH. **(D)** DTX-SSS-Cys. **(E)** Cys-SSS-Cys.

2. The chemical structure of DTX in Figure 2 is suggested to be omitted, and thus the different bridged structures could be shown clearly.

Response: We agree with the reviewer's comments and modified the chemical structure in Figure 2.

Fig. 2. Preparation and characterization of HPNAs. (A) Chemical structures of DTX homodimeric prodrugs. (B) Schematic illustration of the composition of HPNAs. (C) Particle size distribution of HPNAs. (D) Morphology of HPNAs obtained by TEM. Scale bar represents 200 nm. (E) Storage stability of HPNAs at 4°C. (F) Particle size change of HPNAs after co-incubation with fetal bovine serum-containing PBS for 24 h.

3. Generally, the reductive bridged bond is responsive to GSH under physiological

conditions. The authors should investigate the drug release of the prodrug nanoparticles in the solutions with GSH rather than DTT.

Response: We appreciate the reviewer's comments. We evaluated the drug release of the prodrug nanoparticles in the GSH-containing medium. As shown in Supplementary Fig. 28, the release rates of prodrug nanoparticles in 0.1 mM GSH medium were similar to DTT-triggered drug release, so DTT could replace GSH to investigate the reduction responsiveness of prodrugs in vitro. The results were discussed in the revised manuscript (page 15): "We also investigated the drug release of HPNAs in the medium with GSH. As shown in Supplementary Fig. 28, the reduction-responsivity of HPNAs still followed the order of the DTX-STeS-DTX NPs > DTX-SSeS-DTX NPs > DTX-SSS-DTX NPs > DTX-SCS-DTX NPs, which was in a good agreement with the DTT-triggered drug release".

Supplementary Figure 28. In vitro reduction-responsive drug release of HPNAs in the presence of 0.1 mM GSH (n=3).

4. In Figure 4a-f, the drug release profile should be conducted in the condition without DTT and H₂O₂ as control.

Response: We appreciate the reviewer's comments. We also investigated the drug release profile of prodrug nanoassemblies in the medium without DTT or H₂O₂. The results were discussed in the revised manuscript (page 13): "At the same time, we investigated the drug release profile of HPNAs in the medium without redox substances.

The release rates of HPNAs in 24 h were less than 3% (Supplementary Fig. 15)”.

Supplementary Figure 15. In vitro drug release of HPNAs in the medium without redox substance (n=3).

5. In vitro drug release, the mixed solution of PBS and ethanol was chosen as the release medium. The authors should explain the rationality of the choice of release medium.

Response: We appreciate the reviewer’s comments. PBS can provide a buffer environment of pH 7.4. In order to fully dissolve the hydrophobic prodrug, we added 30% ethanol into PBS to meet the sink condition. (*Nat Commun.* 2019, 10, 3211; *Nano Lett.* 2016,16, 5401-5408; *Small.* 2020, 16, e2005039). This part was supplemented in the revised manuscript (page 32): “In release studies, PBS (pH 7.4) containing ethanol (v/v=30%) was used as the medium to meet the sink condition”.

6. In Figure 5A-D, the “DTX” used by the author is confusing. Does it represent DTX solution or Taxotere?

Response: We are sorry to confuse the reviewers. Here DTX stood for Taxotere, and we have corrected it in Fig. 5A-D.

7. The cellular uptake results should be quantified by flow cytometry.

Response: We appreciate the reviewer’s comments. We evaluated the cellular uptake by flow cytometry, and the results were shown in Fig. 5F and Supplementary Fig. 34.

Fig. 5. Cytotoxicity and intracellular bioactivation. (A-D) IC₅₀ values of Taxotere and HPNAs (n=3). (E) Cellular uptake of free coumarin-6 or coumarin-6-labeled HPNAs at 2 h. Scale bar represents 10 μ m. (F) Results of cell uptake by flow cytometry. (G) Illustration of intracellular drug release. (H) Free DTX released from HPNAs after

incubation with 4T1 cells for 72 h (n = 3). **(I) Inhibitory effect of Taxotere and HPNAs on tubulin.** Scale bar represents 20 μm . * $P < 0.05$, ** $P < 0.01$, *** $P < 0.001$, and **** $P < 0.0001$ by two-tailed Student's t-test.

Supplementary Figure 34. (A) Cellular uptake of free coumarin-6 or coumarin-6-labeled HPNAs at 0.5 h. Scale bar represents 10 μm . (B) Results of cell uptake by flow cytometry. **** $P < 0.0001$ by two-tailed Student's t-test.

8. In Figure 5, inhibition of tumor cells and normal cell by prodrug nanoassemblies was examined. Meanwhile, the difference in the redox state of cells needs to be investigated.

Response: We agree with the reviewer's comments, and the redox state of different cells was investigated. The results showed that redox heterogeneity existed among different cells. This part was discussed in the revised manuscript (page 17): "Compared with the normal cells (3T3 cells), the levels of both ROS and GSH in tumor cells (4T1, Hepa 1-6, and B16F10 cells) were higher (Supplementary Fig. 33 A-C). The hybrid bond HPNAs with redox dual-responsiveness could effectively respond to the heterogeneous tumor microenvironment".

Supplementary Figure 33. Intracellular ROS and GSH levels of the 3T3, Hepa 1-6, 4T1, and B16F10 cells. **(A)** Flow cytometry analysis for intracellular ROS of the different cells. **(B)** Histogram of flow analysis for intracellular ROS. **(C)** Intracellular GSH concentrations of the 3T3, Hepa 1-6, 4T1, and B16F10 cells. *** $P < 0.001$ and **** $P < 0.0001$ by one-way ANOVA.

9. The microtubule polymerization assay is recommended to verify cytotoxicity.

Response: We appreciate the reviewer's comments. We investigated the inhibition of microtubules of Taxotere and HPNAs in 4T1 cells. The tubulin in cells was labeled with red fluorescent probe, and the higher red fluorescent intensity indicated the stronger inhibition ability of tubulin depolymerization. The results showed that the order of inhibition of microtubule depolymerization was Taxotere > DTX-STeS-DTX NPs > DTX-SSeS-DTX NPs > DTX-SSS-DTX NPs > DTX-SCS-DTX NPs, which was consistent with the cytotoxicity assays.

In addition, according to the editor's suggestion, we evaluated the apoptosis of 4T1 cells to cross-verify the cytotoxicity. Compared with other HPNAs, DTX-STeS-DTX

NPs could induce the most extensive apoptosis because of its more effective intracellular release. The results were discussed in the revised manuscript (page 17): “In addition, compared with other HPNAs, DTX-STeS-DTX NPs showed the strongest inhibitory effect on tubulin and elicited the most extensive apoptosis, which was also in an agreement with the intracellular release (Fig. 5I, Supplementary Fig. 36, 37). The apoptosis rates of Taxotere and HPNAs were as follows: Taxotere (52.2%) > DTX-STeS-DTX NPs (43.15%) > DTX-SSeS-DTX NPs (37.04%) > DTX-SSS-DTX NPs (32.3%) > DTX-SCS-DTX NPs (25.8%)”.

Fig. 5. Cytotoxicity and intracellular bioactivation. (A-D) IC₅₀ values of Taxotere and HPNAs (n=3). (E) Cellular uptake of free coumarin-6 or coumarin-6-labeled HPNAs at 2 h. Scale bar represents 10 μm. (F) Results of cell uptake by flow cytometry. (G) Illustration of intracellular drug release. (H) Free DTX released from HPNAs after

incubation with 4T1 cells for 72 h (n = 3). **(I) Inhibitory effect of Taxotere and HPNAs on tubulin.** Scale bar represents 20 μm . * $P < 0.05$, ** $P < 0.01$, *** $P < 0.001$, and **** $P < 0.0001$ by two-tailed Student's t-test.

Supplementary Figure 36. The fluorescence intensity in CLSM images analyzed by Image J. (n = 3). ** $P < 0.01$, *** $P < 0.001$ and **** $P < 0.0001$ by one-way ANOVA.

Supplementary Figure 37. Cellular apoptosis assay of Taxotere and HPNAs in 4T1 cells.

10. The discussion part of the manuscript needs to be improved. It is suggested that the author objectively analyze the limitations of the research and the problems that need to be solved in the process of clinical transformation.

Response: We appreciate the reviewer's comments. There are still some issues to be solved to realize the drug delivery systems from bench to bedside. This part was discussed in the revised manuscript (page 27): "In the process of the basic idea to the clinical trial, HPNAs suitable for industrial production and toxicity related to selenium and tellurium need to be paid much attention. Specifically, the final administration form of HPNAs is necessary to be determined (lyophilized powder or solution). In addition, the chemical and physical stability need to meet the requirements of production and sterilization. Moreover, it might be necessary to closely monitor the toxicity related to selenium and tellurium elements in clinical trials".

Reviewer 3:

In this manuscript, the hybrid chalcogen bond was creatively introduced to the prodrug nanoassemblies, and its effect on the self-assembly, redox-responsivity and antitumor efficacy of DTX dimeric prodrugs was investigated in comparison with the previous trisulfide bond. The authors evaluated the self-assembly ability, bioactivation, pharmacokinetic behavior, biodistribution, and pharmacodynamics, and gained in-depth insight into the advantages of hybrid chalcogen bond on the dimeric prodrug nanoassemblies to address heterogeneous tumor redox-microenvironment. Overall, this study is innovative and the data is interesting. However, the following aspects should be addressed.

1. The prodrug nanoassemblies have been further loaded into DSPE-PEG2K for in vivo application, and this loading strategy may compromise “the high drug loading” advantage of dimer prodrug. Could the author give more explanation for the rationale of this design?

Response: We appreciate the reviewer’s comments. In HPNAs, DSPE-PEG_{2K} does not perform as a carrier, as the prodrugs can self-assemble without DSPE-PEG_{2K}. In our previous study, non-PEGylated prodrug nanoassemblies showed poor stability in PBS and could be easily phagocytosed by the reticuloendothelial system (RES), leading to poor pharmacokinetic behavior and antitumor efficiency (*Nano Lett.* 2014, 14, 5577–5583; *Nano Lett.* 2016, 16, 5401-5408; *Small.* 2016, 12, 6353-6362). Because of the steric hindrance and hydrophilic corona provided by PEG materials, PEG-modified NPs significantly improve the half-life of drugs in blood circulation by avoiding the opsonization effect. Therefore, DSPE-PEG_{2K} was used for PEGylation modification to achieve long systemic circulation of NPs in vivo. What’s more, the proportion of DSPE-PEG_{2K} was only 20% (w/w). Therefore, even with the DSPE-PEG_{2K} modification, the drug loading (over 50%) of HPNAs is still higher than traditional nanomedicine.

2. The storage stability of nanoassemblies for a longer period of time needs to be investigated.

Response: We agree with the reviewer’s comments. The longer period stability was examined. The HPNAs could remain stable for a month. This part of the results has been updated in Fig. 2E.

Fig. 2. Preparation and characterization of HPNAs. (A) Chemical structures of DTX homodimeric prodrugs. (B) Schematic illustration of the composition of HPNAs. (C) Particle size distribution of HPNAs. (D) Morphology of HPNAs obtained by TEM. Scale bar represents 200 nm. (E) Storage stability of HPNAs at 4°C. (F) Particle size change of HPNAs after co-incubation with fetal bovine serum-containing PBS for 24

h.

3. In Figure 4, compared with reduction conditions, prodrug nanoassemblies need more oxidative stimuli to release drugs, which the authors need to explain. In addition, the degradation of prodrugs should be investigated to prove the oxidative response of hybrid bonds.

Response: Being attacked by reducing substances, the DTX dimeric prodrugs produced more hydrophilic intermediate—DTX-SH. In contrast, oxidation intermediates had little change in the prodrug structure, which was still a dimer. Therefore, the release rates under oxidative stimuli were relatively low. According to the reviewer's suggestion, we investigated the degradation of HPNAs under oxidation conditions, and the DTX-STeS-DTX NPs showed the highest oxidation response rate (Supplementary Fig. 16).

Supplementary Figure 16. In vitro the response rate of HPNAs under 1 mM H₂O₂ (n=3).

4. The heterogeneity of redox levels in different cell lines needs to be investigated. The authors should supplement the data of ROS and GSH content in different cells.

Response: We agree with the reviewer's comments, and the redox state of different

cells was investigated. The results showed that redox heterogeneity existed among different cells. This part was discussed in the revised manuscript (page 17): “Compared with the normal cells (3T3 cells), the levels of both ROS and GSH in tumor cells (4T1, Hepa 1-6, and B16F10 cells) were higher (Supplementary Fig. 33 A-C). The hybrid bond HPNAs with redox dual-responsiveness could effectively response to the heterogeneous tumor microenvironment”.

Supplementary Figure 33. Intracellular ROS and GSH levels of the 3T3, Hepa 1-6, 4T1, and B16F10 cells. (A) Flow cytometry analysis for intracellular ROS of the different cells. (B) Histogram of flow analysis for intracellular ROS. (C) Intracellular GSH concentrations of the 3T3, Hepa 1-6, 4T1, and B16F10 cells. *** $P < 0.001$ and **** $P < 0.0001$ by one-way ANOVA.

5. Selenium and tellurium, as microelements in human body, are involved in the regulation of redox balance. Therefore, the author needs to consider whether the hybrid chalcogen bond containing prodrug nanoassemblies will affect the redox level of tumor cells.

Response: We appreciate the reviewer’s comments. We evaluated the effect of

HPNAs on the redox state of tumor cells and discussed it in the revised manuscript (page 18): “Sulfur/selenium/tellurium-containing HPNAs can affect the redox equilibrium of tumor cells while undergoing redox response^{42,43}. It has been found that tumor cells are more sensitive to the redox equilibrium than normal cells^{26,44}. Thus, changes in ROS levels and GSH/GSSG ratio in 4T1 cells after HPNAs treatment were examined. Compared with the control group, DTX-STeS-DTX NPs, DTX-SSeS-DTX NPs, and DTX-SSS-DTX NPs significantly increased intracellular ROS levels and decreased GSH/GSSG (Supplementary Fig. 38-39). These results indicated an increased level of intracellular oxidation, which would further accelerate drug release and tumor cell apoptosis”.

Supplementary Figure 38. Intracellular ROS levels of the 4T1 after treated with Taxotere or HPNAs. **(A)** CLSM images. **(B)** Fluorescence quantitative results of CLSM images (n=3). **(C)** Flow cytometry analysis for intracellular ROS (n=3). * $P < 0.05$, ** $P < 0.01$, *** $P < 0.001$, and **** $P < 0.0001$ by two-tailed Student's t test.

Supplementary Figure 39. Flow cytometric analysis for intracellular GSH/GSSG levels of the 4T1 after treated with Taxotere or HPNAs. **(A)** The concentration of GSH (n=3). **(B)** The concentration of GSSG (n=3). **(C)** The concentration ratio of GSH and GSSG (n=3). * $P < 0.05$, ** $P < 0.01$, and *** $P < 0.001$ by two-tailed Student's t test.

6. Details need to be provided about how IC50 values and tumor volume were calculated.

Response: We appreciate the reviewer's comments. The calculation method of IC₅₀ was added to the revised manuscript (page 33): "The IC₅₀ values were calculated by GraphPad Prism 8, using molar concentration and cell viability ratio as parameter.". GraphPad Prism can easily fit a dose-response curve to determine the IC₅₀ as described on www.graphpad.com. Also, the calculation method of tumor volume was added to the revised manuscript (page 38): "The tumor volume was calculated as follows: Tumor volume (mm³) = (Length × Width × Width)/2".

7. The writing needs further polishing. I suggest that the author standardize the use of technical term to avoid misunderstanding to readers. For example, whether "Mean Diameter" and "Size" mean the same meaning.

Response: We appreciate the reviewer's comments. "Size" and "Mean Diameter" have the same meaning. To avoid misleading, the "Size" in Fig. 2F was revised to "Mean Diameter".

8. The author should pay attention to some details, such as the missing space between ordinate name and unit in Figure 2E.

Response: We appreciate the reviewer's comments. We have carefully checked the manuscript, and the space between ordinate name and unit in Fig. 2E was added.

Reviewers' Comments:

Reviewer #1:

Remarks to the Author:

The authors have responded well to all the reviewer comments. Publication is now recommended subject to the usual typographic and ethics checks

Reviewer #2:

Remarks to the Author:

The authors have properly addressed the reviewers' questions. I believe that the manuscript is now acceptable in the current form in Nature Communications.

Manuscript ID: NCOMMS-22-08741B

Title: Hybrid Chalcogen Bonds in Prodrug Nanoassemblies Provides Dual Redox-Responsivity in the Tumour Microenvironment

Dear reviewers,

Firstly, we are truly grateful to your kind letter and hard work. Reviewers' constructive comments are all valuable and helpful for improving our article. We appreciate the positive comments from the reviewers and revise the manuscript according to the editorial policies and formatting requirements. We hope the revised version is improved to merit further consideration of publication.

Your sincerely

Jin Sun, Ph.D.

Professor, Department of Pharmaceutics

Wuya College of Innovation

Shenyang Pharmaceutical University, 110016, PR China

Tel: +86-24-23986325; Fax: +86-24-23986325

Response to Reviewers' comments:

Reviewer 1:

Comments: The authors have responded well to all the reviewer comments. Publication is now recommended subject to the usual typographic and ethics checks

Response: We feel great thanks for your positive comments and valuable suggestions to improve the quality of our manuscript.

Reviewer 2:

Comments: The authors have properly addressed the reviewers' questions. I believe that the manuscript is now acceptable in the current form in Nature Communications.

Response: We sincerely appreciate your positive comments and professional review work on our article.